# Hierarchical Adaptive Value Estimation for Multi-modal Visual Reinforcement Learning

**Yangru Huang**[1]**, Peixi Peng**[2,3] *****, Yifan Zhao**[1]**, Haoran Xu**[3,4]**,**
**Mengyue Geng**[1]**, Yonghong Tian**[1,2,3] *****

[1]School of Computer Science, Peking University
[2]School of Electronic and Computer Engineering, Shenzhen Graduate School, Peking University
[3]Peng Cheng Laboratory
[4]School of Intelligent Systems Engineering, Sun Yat-sen University
`yrhuang@stu.pku.edu.cn,`
`{pxpeng, zhaoyf, mygeng, yhtian}@pku.edu.cn,xuhr9@mail2.sysu.edu.cn`

## Abstract

Integrating RGB frames with alternative modality inputs is gaining increasing traction in many vision-based reinforcement learning (RL) applications. Existing multi-modal vision-based RL methods usually follow a Global Value Estimation (GVE) pipeline, which uses a fused modality feature to obtain a unified global environmental description. However, such a *feature-level* fusion paradigm with a single critic may fall short in policy learning as it tends to overlook the distinct values of each modality. To remedy this, this paper proposes a Local modality-customized Value Estimation (LVE) paradigm, which dynamically estimates the contribution and adjusts the importance weight of each modality from a *value-level* perspective. Furthermore, a task-contextual re-fusion process is developed to achieve a *task-level* re-balance of estimations from both feature and value levels. To this end, a **Hierarchical Adaptive Value Estimation (HAVE)** framework is formed, which adaptively coordinates the contributions of individual modalities as well as their collective efficacy. Agents trained by HAVE are able to exploit the unique characteristics of various modalities while capturing their intricate interactions, achieving substantially improved performance. We specifically highlight the potency of our approach within the challenging landscape of autonomous driving, utilizing the CARLA benchmark with neuromorphic event and depth data to demonstrate HAVE's capability and the effectiveness of its distinct components. The code of our paper can be found at https://github.com/Yara-HYR/HAVE.

## 1 Introduction

Recent years have witnessed a renewed interest in multi-modal perception in the computer vision research community [10, 36, 20, 26, 37]. For many visual tasks such as semantic segmentation and object detection, the inclusion of multi-modal data (*e.g.*, depth, infrared) is proven to be indisputably beneficial [41, 29, 49, 25]. This trend equally applies to the intelligent agents in vision-based Reinforcement Learning (RL), in which multi-modal inputs can also promote decision robustness [39, 19, 1, 11, 31]. For instance, an autonomous-driving agent taking RGB frames solely as input may frequently suffer from extreme light conditions, as shown in Fig. 1(a). Nevertheless, combining additional sensory inputs such as event signals coming from a neuromorphic event camera [27] can effectively alleviate these problems, enabling a more comprehensive realization of traffic status [5, 32, 12].

37th Conference on Neural Information Processing Systems (NeurIPS 2023).

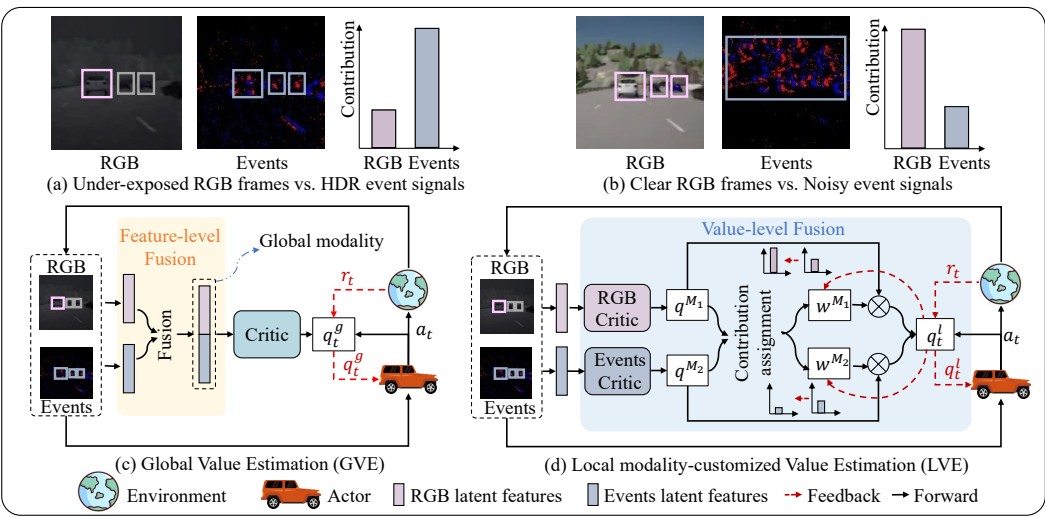

Figure 1: Top: Desired modality contributions of (a) under-exposed RGB frame vs. high dynamic range (HDR) event signals, and (b) clear RGB frame vs. event signals saturated by background noise. Bottom: Multi-modal fusion at (c) the feature level, where only the fused modality feature is used for value evaluation, and (d) the value level, where individual value estimation is conducted for each modality to identify which one performs better under the current circumstance.

Despite the abundance of fruitful studies in the field of multi-modal visual perception, a majority of them focus on traditional static learning tasks. In contrast, the dynamic and unlabeled nature of RL renders the development of multi-modal agents exceedingly difficult [6]. To reconcile multi-sensory data, existing methods [18, 36, 6, 22, 40] train the agents with fused modality features and form a mixed global modality, as shown in Fig.1(c). The policy is then learned via Global Value Estimation (GVE), in which the global modality feature is responsible for environment description and then paired with action to compute a single Q-value. Although being an effective and widely adopted paradigm, utilizing GVE alone may overlook the distinct task-related contributions of different modalities. Specifically, due to the diverse attributes of sensors, single modalities may not contribute equally under multiple environmental conditions (*e.g.*, the cases given in Fig. 1 (a) and (b)). The *feature-level* fusion process in GVE, no matter how sophisticated, inevitably disregards the unique value of each sensory data since only a single critic is shared over multiple modalities. As a result, negative interference between modalities may occur and compromise learning performance.

Based on the above analysis, we propose *Local modality-customized Value Estimation (LVE)*, as illustrated in Fig. 1 (d). Different from feature-level fusion, LVE learns a single policy with distinct, per-modality value calculation, forming a *value-level* fusion paradigm through a tailor-designed Q-value weighting process. As a result, the modality-specific contributions can be explicitly estimated, which promotes policy flexibility. In addition, while LVE is intrinsically a better alternative than GVE, the two paradigms are not competitive but complementary. Considering this, we further develop a *task-contextual re-fusion* process, which utilizes an efficient fusing network guided directly by the task reward to reach a *task-level* balance between LVE and GVE. The collaboration of the above components eventually forms a **Hierarchical Adaptive Value Estimation (HAVE)** framework for multi-modal vision-based RL, yielding a potent policy that can flexibly leverage the unique strengths of each modality while profiting from the comprehensive information from all modalities. We specifically highlight the potency of our approach to the challenging autonomous driving task. In particular, we explicitly consider the neuromorphic event camera [27] signals that capture the motion information of the environment, which well-suites our task but has not yet been explored by existing multi-modal RL algorithms.

In summary, the contributions of our work are three-fold: 1) We design a novel hierarchical adaptive value estimation (HAVE) framework for multi-modal vision-based RL. HAVE distinctively features a local modality-customized value estimation (LVE) paradigm to enable optimized reward allocation based on modality importance. 2) We develop a task-contextual re-fusion process to merge the profi-

ciencies of LVE and GVE, allowing HAVE to benefit from both particularized and unified modality values. 3) Our approach achieves state-of-the-art performance on challenging autonomous driving tasks. As the first multi-modal RL algorithm considering event camera signals as one of the input modalities, our approach exhibits superior performance under various environmental conditions, demonstrating its potential as an effective multi-modal vision-based RL solution.

## 2 Related Work

**Vision-based RL** aims to train agents that receive raw image-based observations from environments for decision-making. Compared to state-based RL, vision-based RL has found significant use in many practical tasks, from video game playing [34] to robotic manipulation [21]. However, learning policy directly from such high-dimensional input is challenging. To tackle the problem, a considerable amount of works have been developed, including 1) applying data augmentation to increase the diversity of samples [50, 24, 30], 2) introducing auxiliary tasks such as contrastive loss [23, 54, 2], 3) pretraining an encoder to improve the representational ability [51, 28, 42], and 4) modeling environment dynamics in the latent space [17, 16, 38]. Although most vision-based RL methods adopt RGB camera frames as inputs, some recent works have also started to explore new sensors and data formats for RL, such as event cameras [46, 47] which captures fast and asynchronous light changes with high temporal precision.

**Multi-modal Visual Learning** has been extensively studied in the field of computer vision [3]. With the development of sensor technologies, it becomes effortless to acquire sufficient data from complemented visual modalities (*e.g.*, RGB, infrared, depth, and event signals). As a result, many multi-modal learning methods are proposed for traditional visual tasks, such as object detection [29] and segmentation [10, 25]. For multi-modal RL, the agent's observation space is modified to include all modalities. Recent works have started to focus on multi-modal vision-based RL due to its improved effectiveness compared with using only single-modality data. Chen *et al.*propose a multi-modal state-space model trained with mutual information lower-bound to promote the consistency between the latent codes of each modality [6]. A fusion network is proposed by Khalil *et al.*to produce accurate joint multi-modal perception and motion prediction for autonomous driving [22]. Ma *et al.*propose a multi-modal RL approach that focuses on modality alignment and importance enhancement [31]. There are also multi-modal RL methods designed for other tasks such as robot control [4] and dialog system [33, 53]. Despite their distinct technical details, most of the existing methods perform a feature-level fusion of modalities and ignore the properties of RL itself. In fact, policy learning depends on the value estimation of the critic. Consequently, we view the multi-modal visual RL problem from a novel adaptive value estimation perspective. Instead of concentrating solely on robust global modality features, our approach synergistically balances individual modalities' contributions, leading to a more equitable and efficient value allocation. Leveraging the task-contextual re-fusion mechanism, our method further capitalizes on both feature-based and value-based fusion paradigms, resulting in a more robust policy.

## 3 Methodology

### 3.1 Preliminaries

**Problem Definition** We formulate the task of multi-modal vision-based RL as a Markov Decision Process (MDP) [44] with multiple observations, which is defined by a tuple $(\mathcal{S}, \mathcal{A}, \mathcal{P}, \mathcal{R}, \gamma)$, where $\mathcal{S} = \prod_{i=1}^{d} \mathcal{O}^{M_i}$ is the joint observation space of $d$ modalities and $\mathcal{O}^{M_i}$ is the observation space of modality $i$. $\mathcal{A}$ is all possible actions the agent can take. $\mathcal{P} : \mathcal{S} \times \mathcal{A} \times \mathcal{S} \rightarrow [0, 1]$ is the transition probability function, $\mathcal{P}(o_{t+1}|o_t, a_t)$ denotes the probability of transitioning from joint observation $o_t = (o_t^{M_1}, o_t^{M_2}, \ldots, o_t^{M_d})$ at time step $t$ to the next joint observation $o_{t+1} = (o_{t+1}^{M_1}, o_{t+1}^{M_2}, \ldots, o_{t+1}^{M_d})$ after taking action $a_t$ from a policy function $\pi$.

**Soft Actor-Critic (SAC)** Our approach is built on SAC [14, 15], which goal is to learn a policy that maximizes the expected cumulative reward while maintaining exploration by encouraging diverse actions. In SAC, the objective function is given by introducing a policy entropy term:

$$\mathcal{L}_\pi = \mathbb{E}_{a_t \sim \pi} \left[ Q(o_t, a_t) - \alpha \log \pi(a_t|o_t) \right], \tag{1}$$

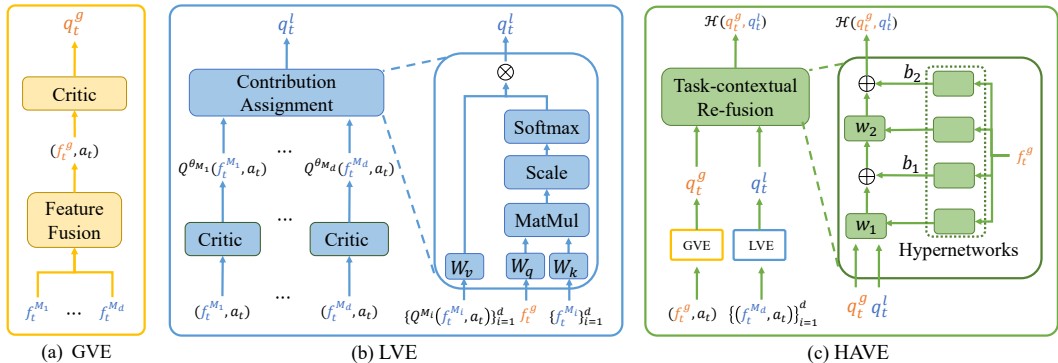

Figure 2: HAVE framework and its distinct components. (a) Global Value Estimation (GVE), (b) Local modality-customized Value Estimation (LVE), and (c) HAVE with task-level value re-fusion.

where $Q$ is the value function, and $\alpha$ is a temperature parameter that controls the trade-off between exploration and exploitation. The value function is trained using the Bellman equation and a soft Q-function update:

$$\mathcal{L}_Q = \mathbb{E}_{(o_t, a_t) \sim \mathcal{D}} \left[ \left( Q(o_t, a_t) - \left( \mathcal{R}(o_t, a_t) + \gamma \mathbb{E}_{o_{t+1} \sim p} \left[ V(o_{t+1}) \right] \right) \right)^2 \right], \tag{2}$$

where $o_t$ and $a_t$ are sampled from the replay buffer $\mathcal{D}$ and $\gamma$ is the discount factor. $V(o_{t+1})$ is the soft state value function, defined as:

$$V(o_{t+1}) = \mathbb{E}_{\tilde{a}_{t+1} \sim \pi} \left[ \bar{Q}(o_{t+1}, \tilde{a}_{t+1}) - \alpha \log \pi(\tilde{a}_{t+1} | o_{t+1}) \right], \tag{3}$$

where $\bar{Q}$ denotes an exponential moving average of the critic network $Q$ and $\tilde{a}_{t+1}$ comes from the current policy.

**Event Cameras and Representations** Despite RGB frames, another major modality that we use to evaluate our approach is the event signals generated by a neuromorphic event camera [27]. Each pixel in the event camera outputs a positive/negative event signal whenever the log light intensity of that pixel has increased/decreased by a constant threshold. The event signals have an extremely high dynamic range (up to 120 dB) and can reach a high temporal resolution in the order of $\mu$s. Therefore, they are able to capture the missing motion clues that are missed in the RGB frames for many visual tasks such as autonomous driving and robot navigation. To utilize event signals, we use the stacking based on time (SBT) [48] representation that splits the event sequence into fixed temporal bins, forming an event frame representation with multiple channels similar to RGB frames.

### 3.2 Global Value Estimation

In Global Value Estimation (GVE), a total of $d$ observation encoders are used to extract observation features $f_t^{M_i}$ for each modality observation $o_t^{M_i}$ at time step $t$. A global modality feature $f_t^g$ is then calculated by fusing all modality features. For simplicity, we directly concatenate $f_t^{M_i}(i = 1, 2, \ldots, d)$ along the channel dimension. Given $f_t^g$, GVE first computes the value estimation $q_t^g = Q^{\theta_g}(f_t^g, a_t)$ using a single global critic network $Q^{\theta_g}$, as shown in Fig. 2(a). Following the SAC algorithm, the target value $y_t^g$ is then calculated as:

$$y_t^g = \mathcal{R}(o_t, a_t) + \gamma V(f_{t+1}^g), \tag{4}$$

where $\mathcal{R}(o_t, a_t)$ is the reward returned by the environment. The soft state value function $V(f_{t+1}^g)$ is denoted as:

$$V(f_{t+1}^g) = \mathbb{E}_{\tilde{a}_{t+1} \sim \pi} \left[ \bar{Q}^{\theta_g}(f_{t+1}^g, \tilde{a}_{t+1}) - \alpha \log \pi(\tilde{a}_{t+1} | f_{t+1}^g) \right]. \tag{5}$$

However, since using a single value function $Q^{\theta_g}$ shared over all modalities, it is hard to dynamically balance the relative contribution of each $f_t^{M_i}$ under complex environmental conditions.

### 3.3 Local Modality Customized Value Estimation

The objective of Local modality-customized Value Estimation (LVE) is to effectively differentiate and quantify the unique contribution of each modality, thereby facilitating a more granular and modality-aware decision-making process. Specifically, in LVE, $d$ individual value functions $Q^{\theta_{M_1}}, Q^{\theta_{M_2}}, ..., Q^{\theta_{M_d}}$ are setup instead of global-only functions in GVE. These value functions share the same network architecture but are learned separately. To facilitate value decomposition, an assignment network is designed to assimilate all estimated values and re-calibrate them according to the collective environmental condition portrayed by modality features.

**Value Inference:** We assume that the locally-aggregated total value $q_t^l$ can be approximately decomposed into a linear combination of the separate value functions across different modalities under a shared policy:

$$q_t^l \approx \sum_{i=1}^{d} w_t^{M_i} Q^{\theta_{M_i}}(f_t^{M_i}, a_t), \tag{6}$$

where $w^{M_i}$ denotes the contribution weight of modality $M_i$ to the total value. The value function $Q^{\theta_{M_i}}$ for each modality processes the current individual modal observation feature $f_t^{M_i}$ along with the current action $a_t$, generating the estimated action values at each time step. Each $Q^{\theta_{M_i}}$ is learned by backpropagating gradients from the Q-learning rule. Similarly, the total target value $y_t^l$ can be defined as:

$$y_t^l \approx \mathcal{R}(o_t, a_t) + \gamma V(f_{t+1}^{M_1}, ..., f_{t+1}^{M_d}). \tag{7}$$

The soft state value function $V(f_{t+1}^{M_1}, ..., f_{t+1}^{M_d})$ is then defined as:

$$V(f_{t+1}^{M_1}, ..., f_{t+1}^{M_d}) = \mathbb{E}_{\tilde{a}_{t+1} \sim \pi} \left[ \sum_{i=1}^{d} v_{t+1}^{M_i} \bar{Q}^{\theta_{M_i}}(f_{t+1}^{M_i}, \tilde{a}_{t+1}) - \alpha \log \pi(\tilde{a}_{t+1} | f_{t+1}^g) \right], \tag{8}$$

where $v_t^{M_i}$ is the contribution weight of modality $M_i$ to the the estimated Q-value from $\bar{Q}^{\theta_{M_i}}$ at the next time. $\tilde{a}_{t+1} \sim \pi(\cdot | f_{t+1}^g)$ comes from the current policy according to the global modality feature $f_{t+1}^g$ since there is only one policy network for the multi-modal decision task.

**Contribution Assignment:** Given value decomposition in Eq. 6 and Eq. 7, a critical next step is to calculate accurate modality weights $w_t^{M_i}$ and $v_t^{M_i}$ to ensure precise total value estimation. To achieve this, the contribution assignment process involves modality interactions through an assignment network based on the attention mechanism. Taking the calculation of $w_t^{M_i}$ as an example, the global modality feature $f_t^g$ is used as a bridge to ensure information flow between all individual modalities. As demonstrated in Fig. 2(b), two fully connected (FC) layers with parameters $W_q$ and $W_k$ are used to project both $f_t^{M_i}$ and $f_t^g$ into a $p$-dimensional common latent subspace. Then $w_t^{M_i}$ can be obtained through a softmax function as:

$$w_t^{M_i} = \frac{\exp((W_q f_t^g)(W_k f_t^{M_i})^\top / \sqrt{p})}{\sum_{i=1}^{d} \exp((W_q f_t^g)(W_k f_t^{M_i})^\top / \sqrt{p})}, \tag{9}$$

where $\sqrt{p}$ is used for scaling to prevent vanishing gradients [45]. The computation of $v_{t+1}^{M_i}$ follows a similar derivation, which is omitted here for brevity.

By assigning the value function in a modality-customized manner, LVE adeptly manages multi-modal sensory inputs, enhancing model efficiency and interpretability. Our approach readily adapts to complex environmental scenarios, accommodating varying modality importance levels.

### 3.4 Task-Contextual Re-fusion

**Feature-based *vs.* Value-based Fusion** From the details of GVE and LVE, we see that the main difference distinguishing them is their fusion principle (feature-based vs. value-based). Intuitively, the two paradigms are not competitive but complementary. Specifically, in the context of RL, value-based fusion offers significant advantages over feature-based fusion since the former is more related to the actual decision. In the meantime, the feature-based fusion method can use its collective modality value to offer a value-based one with a sturdy global reference, mitigating inaccuracies caused by input noise. Therefore, we further develop task-contextual re-fusion for a subsequent re-fuse of GVE with LVE in the reward/task level. The idea is to directly bridge the two fusion mechanisms

based on environmental reward feedback, allowing the task to determine which paradigm better suits the current situation.

**Details of the Re-fusion Process** Given the estimated value $q_t^g$ and $q_t^l$ obtained from GVE and LVE, respectively, we adopt a dynamic fusion mechanism with a re-fusion network $\mathcal{H}$ to fuse $q_t^g$ and $q_t^l$. Specifically, $\mathcal{H}$ consists of a series of FC layers and hypernetworks. Each hypernetwork consists of a single linear layer, which receives the global modality feature $f_t^g$ as input and produces a weight matrix or a bias vector for a single FC layer in $\mathcal{H}$. To fuse $q_t^g$ and $q_t^l$, $f_t^g$ is first sent into all hypernetworks to obtain the parameters of the FC layers in $\mathcal{H}$, then $q_t^g$ and $q_t^l$ are sent into $\mathcal{H}$ to get the final total value $\mathcal{H}(q_t^g, q_t^l)$. The final target value can be obtained using a similar way, wherein the estimated Q-value from the target Q-network at the next time is processed by another mixing network $\mathcal{H}'$. For brevity, we denote the this target value as $\mathcal{H}'(y_t^g, y_t^l)$.

### 3.5 Learning Framework

Coping GVE with LVE and task-contextual re-fusion, our approach forms a unified framework, which we name as **Hierarchical Adaptive Value Estimation (HAVE)** for multi-modal vision-based RL. In this section, we elaborate on the training details of our HAVE framework.

**Policy Evaluation with HAVE** For the training of policy evaluation, we minimize the temporal difference (TD) error between the predicted value $\mathcal{H}(q_t^g, q_t^l)$, and the target value $\mathcal{H}'(y_t^g, y_t^l)$ following the SAC algorithm:

$$\mathcal{L}_Q = \mathbb{E}_{(o_t, a_t) \sim \mathcal{D}} \left[ \left( \mathcal{H}(q_t^g, q_t^l) - \mathcal{H}'(y_t^g, y_t^l) \right)^2 \right], \tag{10}$$

where $\mathcal{D}$ denotes the replay buffer.

**Policy Improvement** The policy evaluation described above can cover reasonable Q-values to help find better policies. In the policy improvement step, we leverage the values obtained from GVE, LVE, and the task-contextual re-fusion process to update the policy $\pi$. To be specific, the goal is to maximize the expected cumulative reward by selecting actions that have the highest combined value estimates:

$$\mathcal{L}_\pi = \mathbb{E}_{a_t \sim \pi} \left[ \mathcal{H}(q_t^g, q_t^l) - \alpha \log \pi(a_t | f_t^g) \right]. \tag{11}$$

After identifying the optimal action under the improved policy, we can update the policy parameters by backpropagating gradients.

**Auxiliary Losses** Besides policy evaluation and improvement, we also learn to predict the rewards and next latent states as in DeepMDP [13] to assist representation learning, providing a latent state space that is consistent with environmental dynamics:

$$\mathcal{L}_{aux} = \|\mathcal{P}^g(f_t^g, a_t) - f_{t+1}^g\|^2 + \sum_{i=1}^d \|\mathcal{P}^{M_i}(f_t^{M_i}, a_t) - f_{t+1}^{M_i}\|^2 + [\mathcal{R}^g(f_t^g, a_t) - \mathcal{R}(o_{t+1}, a_{t+1})]^2, \tag{12}$$

where $\mathcal{P}^g$ and $\mathcal{P}^{M_i}(i = 1, 2, \ldots, d)$ are transition networks for the global and individual modalities, respectively. $\mathcal{R}^g$ is the reward prediction network. All these networks are formed by FC layers and a more detailed architecture description can be found in the supplementary material.

**Overall Training Objective** Given the above training objectives, the overall loss of HAVE is finally defined by:

$$\mathcal{L} = \mathcal{L}_Q + \mathcal{L}_\pi + \mathcal{L}_{aux}. \tag{13}$$

By iteratively performing policy evaluation and improvement steps, HAVE can effectively exploit the strengths of both global and distinct value estimates, ultimately converging to an improved policy that can handle the challenges posed by a broad range of environmental modalities.

### 3.6 Further Analyses

Our hierarchical adaptive value estimation framework for multi-modal RL process offers several key benefits, which we list as follows:

**Remark 1 (Prevention of Modality Dominance):** *The mechanism of LVE and task-contextual re-fusion prevents one modality from dominating the others, thereby avoiding the issue of modality collapse.*

To realize this, consider that the mixing network $\mathcal{H}$ in the task-contextual re-fusion performs fixed non-linear transformations on both $q_t^g$ and $q_t^l$ given $f_t^g$, where $q_t^l$ is formed by the convex combination of individual modality values (Eq. 6) as in LVE. Therefore, we have:

$$\text{sign}(\frac{\partial \mathcal{H}(q_t^g, q_t^l)}{\partial q_t^{M_i}}) = \text{sign}(\frac{\partial \mathcal{H}(q_t^g, q_t^l)}{\partial q_t^l} \frac{\partial q_t^l}{\partial q_t^{M_i}}) = \text{sign}(\frac{\partial \mathcal{H}(q_t^g, q_t^l)}{\partial q_t^l}), \forall i \in [1, 2, \ldots, d], \quad (14)$$

where $\text{sign}(\cdot)$ is the real sign function, $q_t^{M_i} = Q^{\theta_{M_i}}(f_t^{M_i}, a_t)$ is the individual modality value and $\frac{\partial q_t^l}{\partial q_t^{M_i}} = w_t^{M_i} > 0$ are the modality weights. Eq. 14 indicates that the contributions of all modalities are either increasing or decreasing together during the learning process, which prevents opposite gradient values between modalities that enhance some of them while suppressing others. In addition, the calculation of modality weights $w_t^{M_i}$ based on softmax global-individual feature similarity (Eq. 9) further prevents modality collapse caused by persistent zero weights.

**Remark 2 (Pareto Optimality of Modalities):** *Assuming continuous action space, the global optimal action produced by our approach achieves a Pareto optimum across individual modalities.*

Suppose the optimal action given some fixed modality features is $a^*$, which implies $\frac{\partial \mathcal{H}(q_t^g, q_t^l)}{\partial a^*} = 0$. Combining Eq. 6 with the chain rule, we have:

$$\frac{\partial \mathcal{H}(q_t^g, q_t^l)}{\partial a^*} = \frac{\partial \mathcal{H}(q_t^g, q_t^l)}{\partial q_t^l} \frac{\partial q_t^l}{\partial a^*} = \frac{\partial \mathcal{H}(q_t^g, q_t^l)}{\partial q_t^l} \sum_{i=1}^{d} w_t^{M_i} \frac{\partial q_t^{M_i}}{\partial a^*} = 0. \quad (15)$$

Considering the non-linear structure of $\mathcal{H}$, $\frac{\partial \mathcal{H}(q_t^g, q_t^l)}{\partial q_t^l}$ is unlikely to be zero for all $q_t^l$. This implies that $\sum_{i=1}^{d} w_t^{M_i} \frac{\partial q_t^{M_i}}{\partial a^*} = 0$ for all $i = 1, 2, \ldots, d$, which suggests that at the point $a^*$, the positive weighted sum of the gradients of individual modalities' values is zero. This implies that there is no alternative action assignment in the vicinity of $a^*$ that can increase the value for any individual modality without decreasing the value for some other modality. Such condition forms a Pareto optimum across the modalities, which indicates the resources (modality values) are allocated in the most efficient way possible [35].

# 4 Experiments

## 4.1 Settings

**Environments** To evaluate our approach under realistic and challenging multi-modal environments, we employ the CARLA simulator [8], which is a widely used open-source platform for autonomous driving research. CARLA provides a rich and realistic urban environment to evaluate autonomous driving agents in various traffic scenarios. Three distinct modalities are adopted: 1) RGB frames, 2) event signals generated by CARLA's synthetic event-camera simulator, and 3) per-pixel depth frames. Eight different weather settings are used to provide thorough coverage of different environmental conditions. The action space consists of continuous control actions, such as steering, acceleration, and braking. Similar to [52, 9], the reward function is designed to encourage the agent to maintain a safe distance from 20 other moving vehicles and obstacles, and drive as far as possible along the highway of CARLA's Town04 map in 1000 time steps. We use the single camera view setting on the vehicle's roof with 60-degree views.

**Implementation Details** Our approach is implemented based on SAC [14, 15] and DeepMDP [13]. The same encoder network architecture and training hyperparameters are adopted for all comparative methods. Following common practices [50, 23], we convert each modality data into its corresponding image-based representations and stack several consecutive images to infer temporal information. The spatial resolution of the input images is $128 \times 128$ and the channel numbers of RGB, event and depth frames are 3, 5 and 1, respectively. All methods are trained for 120k frames across 5 random seeds to report the mean and standard deviation of the rewards. We evaluate the performance of each approach in terms of driving episode reward and distance. Other details are provided in the supplementary material.

| Weather | Measures | SAC | DrQ | DeepMDP | TransFuser | EFNet | Ours-LVE | Ours-HAVE |
|---|---|---|---|---|---|---|---|---|
| ClearNight | ER | 186± 65 | 242± 84 | 225± 87 | 260± 95 | 241± 89 | 274± 68 | **319± 71** |
|  | D(m) | 112± 39 | 169± 41 | 161± 51 | 178± 43 | 170± 62 | 192± 50 | **212± 52** |
| CloudyNight | ER | 218± 71 | 248± 69 | 265± 85 | 280± 71 | 289± 64 | 295± 67 | **322± 96** |
|  | D(m) | 132± 64 | 167± 36 | 183± 55 | 195± 43 | 197± 41 | 213± 44 | **217± 53** |
| HardRainNight | ER | 170± 90 | 261± 91 | 255± 77 | 287± 65 | 275± 72 | 294± 85 | **327± 87** |
|  | D(m) | 107± 63 | 163± 63 | 160± 36 | 204± 56 | 203± 54 | 209± 54 | **229± 51** |
| WetNight | ER | 189± 79 | 234± 97 | 241± 47 | 274± 76 | 290± 93 | 289± 112 | **304± 102** |
|  | D(m) | 127± 57 | 155± 53 | 164± 36 | 171± 58 | 201± 56 | 196± 61 | **222± 69** |
| ClearNoon | ER | 235± 58 | 280± 94 | 269± 64 | 282± 53 | 234± 79 | 294± 82 | **336± 76** |
|  | D(m) | 153± 49 | 195± 40 | 187± 36 | 193± 39 | 150± 46 | 186± 61 | **223± 44** |
| CloudyNoon | ER | 201± 87 | 274± 77 | 226± 24 | 277± 67 | 261± 78 | 293± 76 | **315± 82** |
|  | D(m) | 138± 68 | 170± 42 | 136± 16 | 171± 42 | 164± 58 | 186± 44 | **209± 68** |
| HardRainNoon | ER | 189± 74 | 220± 72 | 248± 59 | 264± 99 | 279± 91 | 287± 95 | **316± 88** |
|  | D(m) | 104± 56 | 129± 60 | 161± 43 | 178± 62 | 208± 76 | 207± 69 | **218± 63** |
| WetNoon | ER | 209± 81 | 245± 83 | 226± 52 | 304± 81 | 273± 82 | 296± 84 | **341± 78** |
|  | D(m) | 136± 64 | 172± 58 | 169± 38 | 213± 51 | 204± 70 | 215± 54 | **239± 55** |

Table 1: Comparison with state-of-the-art methods on eight different kinds of weather. ER denotes episode return and D is distance in meters. The best results are **bolded** and the second best results are underlined.

## 4.2 Comparison with State of the Art

We compare our method with a variety of methods, including the SAC [15] baseline, DeepMDP [13], DrQ [50], TransFuser [7], and EFNet [43]. For RL methods SAC, DrQ and DeepMDP, we directly concatenate the features of the different modalities as the input for subsequent value and policy learning. For TransFuser and EFNet which are designed for traditional visual tasks, we only adopt their advanced modality fusion modules and keep DeepMDP as the RL algorithm for a fair comparison.

**Results of Two Modalities** We first evaluate different methods under two modality inputs (RGB frames and event signals) in Table 1. The results show that our method achieves the highest episode reward and driving distance under all eight weather conditions. DrQ and DeepMDP perform better than the SAC baseline with limited improvement, showing that the feature-fusion based GVE paradigm, together with simple feature concatenation, cannot fully extract the expressive power of each modality. The improved performance of TransFuser and EFNet over DeepMDP reflects the importance of the advanced modality feature fusion mechanism. However, they are still inferior to ours-LVE, which uses the proposed LVE paradigm to explicitly assign modality contributions weights with value-based fusion. The results indicate that the key to multi-modal RL is to consider the suitability of individual modalities. Finally, by using the task-contextual re-fusion process to integrate GVE and LVE, our full method achieve superior performance even with simple feature concatenation fusion, proving the effectiveness of our task-driven hierarchical design.

**Results of Multiple Modalities** We then evaluate our method on three modalities (RGB frames, event signals, and depth frames). Although TransFuser and EFNet technically possible to scale both methods to accommodate more than two modalities, such an adaptation would require significant modifications to the implementation, along with a quadratic increase in computational complexity due to cross attention mechanism. Thus we directly compare our method with SAC, DrQ and DeepMDP. The training curves under two weather conditions are demonstrated in Fig. 3, which also show the advantage of our methods. Additional experiment results can be found in the supplementary material.

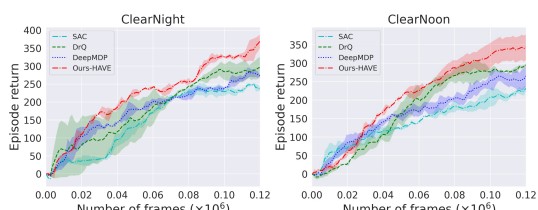

Figure 3: Performance with three modalities.

## 4.3 Performance Analysis

**Ablation Studies** To systematically evaluate the effectiveness of the proposed HAVE approach and its individual components, we conduct a series of ablation experiments on the CARLA benchmark that trains agents using: 1) single modality data with RGB frames or event signals, 2) GVE with feature concatenation, 3) our proposed LVE paradigm, and 4) our full method. The training curves and testing performance are shown in Fig. 4 and Tab. 2, respectively. First, we observe that the indi-

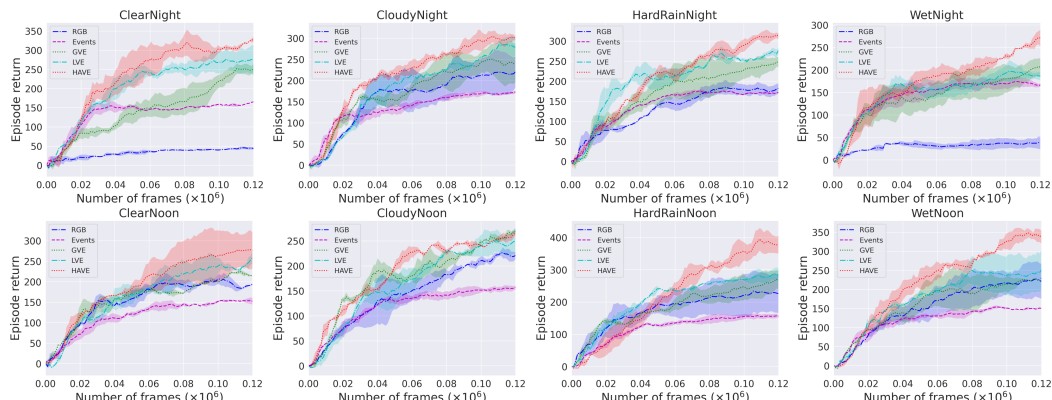

Figure 4: Training curves of individual modalities and different multi-modal training paradigms.

| Methods | Measures | Night | | | | Noon | | | |
|---|---|---|---|---|---|---|---|---|---|
| | | Clear | Cloudy | HardRain | Wet | Clear | Cloudy | HardRain | Wet |
| RGB | ER | $75 \pm 5$ | $194 \pm 36$ | $189 \pm 17$ | $88 \pm 23$ | $214 \pm 55$ | $201 \pm 37$ | $223 \pm 67$ | $232 \pm 74$ |
| | D(m) | $46 \pm 21$ | $121 \pm 24$ | $136 \pm 15$ | $53 \pm 26$ | $125 \pm 42$ | $121 \pm 25$ | $128 \pm 51$ | $141 \pm 54$ |
| Events | ER | $158 \pm 29$ | $187 \pm 32$ | $164 \pm 28$ | $175 \pm 29$ | $145 \pm 37$ | $151 \pm 41$ | $154 \pm 34$ | $147 \pm 40$ |
| | D(m) | $102 \pm 23$ | $129 \pm 26$ | $105 \pm 21$ | $120 \pm 18$ | $86 \pm 27$ | $92 \pm 32$ | $104 \pm 21$ | $86 \pm 36$ |
| GVE | ER | $225 \pm 87$ | $265 \pm 85$ | $255 \pm 77$ | $241 \pm 47$ | $269 \pm 64$ | $226 \pm 24$ | $248 \pm 59$ | $226 \pm 52$ |
| | D(m) | $161 \pm 51$ | $183 \pm 55$ | $160 \pm 36$ | $164 \pm 36$ | $187 \pm 36$ | $136 \pm 16$ | $161 \pm 43$ | $169 \pm 38$ |
| LVE | ER | $\underline{274 \pm 68}$ | $\underline{295 \pm 67}$ | $\underline{294 \pm 85}$ | $\underline{289 \pm 112}$ | $\underline{294 \pm 82}$ | $\underline{293 \pm 76}$ | $\underline{287 \pm 95}$ | $\underline{296 \pm 84}$ |
| | D(m) | $\underline{192 \pm 50}$ | $\underline{213 \pm 44}$ | $\underline{209 \pm 54}$ | $\underline{196 \pm 61}$ | $\underline{186 \pm 61}$ | $\underline{186 \pm 44}$ | $\underline{207 \pm 69}$ | $\underline{215 \pm 54}$ |
| HAVE | ER | $\mathbf{319 \pm 71}$ | $\mathbf{322 \pm 96}$ | $\mathbf{327 \pm 87}$ | $\mathbf{304 \pm 102}$ | $\mathbf{336 \pm 76}$ | $\mathbf{315 \pm 82}$ | $\mathbf{316 \pm 88}$ | $\mathbf{341 \pm 78}$ |
| | D(m) | $\mathbf{212 \pm 52}$ | $\mathbf{217 \pm 53}$ | $\mathbf{229 \pm 51}$ | $\mathbf{222 \pm 69}$ | $\mathbf{223 \pm 44}$ | $\mathbf{209 \pm 68}$ | $\mathbf{218 \pm 63}$ | $\mathbf{239 \pm 55}$ |

Table 2: Test performance of different models in Fig. 4. Table notations are the same as in Table 1.

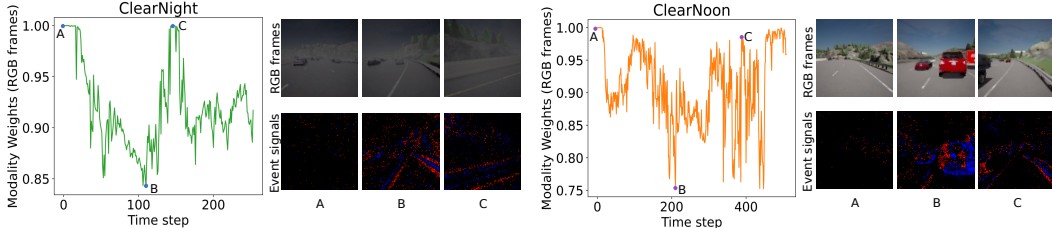

Figure 5: Visualization of modality weights during testing. Note that only weights of RGB frames are drawn and the weights of event signals are one minus RGB frame weights at any time step.

vidual modality is vulnerable to failure under extreme weather conditions. For example, the episode reward trained with RGB frame under ClearNight weather is pretty poor as shown in Fig. 4. Second, directly fusing the two modalities with GVE generally improves performance. However, the performance gain is limited, and in some cases even close to the results of using a single modality (*e.g.*, in WetNight and ClearNoon). This might be caused by the heterogeneity of multi-modal data, which leads to difficulty in determining the target reward. Third, in most cases, LVE performs significantly higher than single modality or GVE-based agents, showing better cooperation of different modality data. Finally, our full method with task-contextual re-fusion clearly outperforms all other models thanks to the synergistic interplay of GVE and LVE at the reward/task-level.

**Visualization** To further obtain an intuitive understanding of our approach, we visualize the modality weights obtained by HAVE in Fig. 5 using RGB frames and event signals under two different weather conditions. The following observations can be made: 1) at the beginning, all vehicles have zero initial speed. As a result, no valid event signal is generated and the weights of RGB modality are nearly 1.0. 2) After speeding up, the weights of event signals are continuously increasing, especially when there are many moving vehicles ahead. 3) However, when the agent is facing complex background (such as road fences and clear roads without close vehicles), the event signals mainly consist of background noise, and the weights of RGB frames will raise again. The varying modality weights show that our approach can indeed adjust modality contributions under different situations, which serves as a key advantage over previous multi-modal RL methods.

# 5 Conclusion and Limitation

We have studied the representational capacity of value functions in multi-modal vision-based RL problems. We hypothesize that the limitation of feature-level fusion methods may come from unclear contributions for different modalities under a single value function. To mitigate this, we have presented a novel Hierarchical Adaptive Value Estimation (HAVE) framework to reconcile both feature-level and value-level fusion in a task/reward-driven manner. Our approach represents one of the first explorations of modality-specific and hierarchical value estimation for multi-modal vision-based RL tasks. Extensive experiment results demonstrate the effectiveness of our approach. However, one limitation of our work is that we mainly focus on the value estimation of multiple visual modalities, while the effectiveness of other forms of modalities (*e.g.*, audio, text) is not verified. In our future work, we will consider utilizing both visual and other modalities, forming a more generalized multi-modal RL framework.

# 6 Acknowledgement

The study was funded by the Key-Area Research and Development Program of Guangdong Province with contract No. 2021B0101400002, the National Natural Science Foundation of China under contracts No. 62027804, No. 61825101, No. 62088102 and No. 62202010, and the major key project of the Peng Cheng Laboratory (PCL2021A13). Computing support was provided by Pengcheng Cloudbrain.

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
