# Supplementary Material for "Hierarchical Adaptive Value Estimation for Multi-modal Visual Reinforcement Learning"

**Yangru Huang**[1], **Peixi Peng**[2,3] *, **Yifan Zhao**[1], **Haoran Xu**[3,4],
**Mengyue Geng**[1], **Yonghong Tian**[1,2,3] *

[1]School of Computer Science, Peking University
[2]School of Electronic and Computer Engineering, Peking University
[3]Peng Cheng Laboratory
[4]School of Intelligent Systems Engineering, Sun Yat-sen University
yrhuang@stu.pku.edu.cn,
{pxpeng, zhaoyf, mygeng, yhtian}@pku.edu.cn,xuhr9@mail2.sysu.edu.cn

The contents of this supplementary material are organized as follows:

Section A provides additional experimental results, including more results with three modalities, performance under dynamic weathers, performance under several challenging or extreme environmental conditions (*e.g.*, increased number of vehicles and dazzling sunlight), results on DeepMind Control Suit, and ablation study of auxiliary losses and the design of re-fusion.

Section B provides further discussions related to our approach. This includes a comparison between value-level dynamic fusion and feature-level dynamic fusion supported by empirical results, the advantages of hierarchical bi-level fusion over uni-level fusion, and the relationship and differences between our approach and the value decomposition techniques in multi-agent RL.

Section C describes the details of the experimental setup, including network architectures, hyperparameters, and hardware details.

Section D states the potential negative societal impacts of our work.

## A  Additional experimental results

### A.1  Experimental Results on Dynamic Weathers

In this experiment, we manipulate the weather conditions in CARLA to simulate smooth changes in the position of the sun over time, while also introducing occasional stormy weather. This manipulation aims to replicate realistic weather transitions that occur naturally, such as the shift from day to night or the transition from sunny to rainy conditions. By incorporating these weather variations, we seek to create a more immersive and dynamic environment that closely resembles real-world weather changes. As shown in Fig. A1, upon observing real-time weather fluctuations, we find that reliance solely on LVE for value fusion resulted in subpar performance. This outcome emphasizes the necessity of feature interaction or feature fusion to tackle intricate situations. Furthermore, an amalgamation of feature fusion and value fusion can offer better performance. This indicates that the feature-level and value-level integration allows for a better understanding of the environment and optimizes decision-making processes in the face of dynamic and challenging scenarios.

37th Conference on Neural Information Processing Systems (NeurIPS 2023).

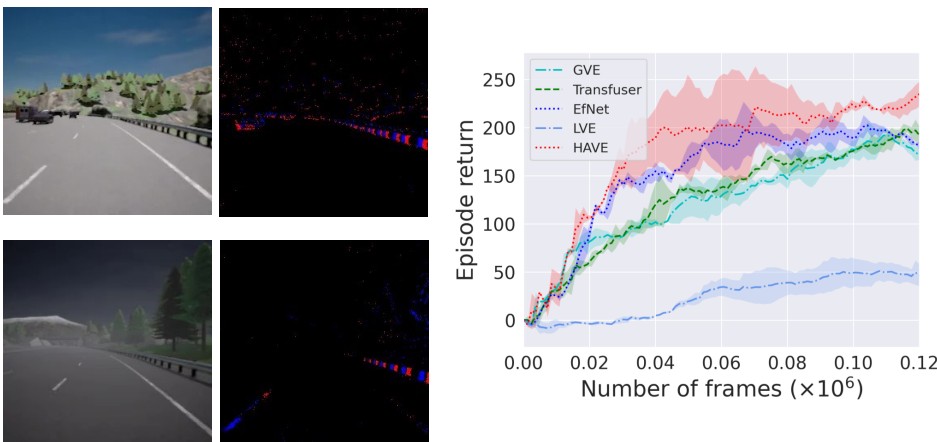

Figure A1: Visualization of dynamic weather changes and corresponding experimental results.

## A.2 Experimental Results on Challenging and Extreme Conditions

**Increased number of vehicles** To enhance the difficulty and assess the effectiveness of our method under more complex road conditions, we increase the number of cars on the road from 20 to 30. This adjustment allows us to evaluate the robustness and adaptability of our approach in handling a larger number of vehicles in the environment. As we increase the number of vehicles on the road, Fig. A2 (a) clearly indicates that HAVE consistently delivers the highest performance. This observation underscores the effectiveness of incorporating both feature fusion and value fusion in our approach. By estimating values from two perspectives (GVE and LVE), our model can successfully capture and leverage the complementary information from multiple modalities, leading to improved performance in scenarios with a higher density of moving targets.

**Extremely Dazzling Sunlight** A particularly challenging scenario in driving occurs when we are moving toward the sun (*e.g.*, driving west in the afternoon and facing the setting sun). In such situations, the intense brightness emitted by the sun can cause overexposure in the RGB camera, leading to loss of details and difficulties in capturing accurate images. As shown in Fig. A2 (b), HAVE gives the best experimental results under dazzling sunlight weather. One potential explanation could be the robustness of the proposed HAVE in handling high variations in scene illumination. That is, HAVE is able to adaptively adjust the contribution of each modality according to its specific value.

## A.3 Additional Experimental Results with Multiple Modalities

Here we provide an additional evaluation of our approach with multiple modalities, including RGB, event signals, and depth. By examining the results in Fig. A3 (a) and Fig. A3 (b), we can obtain more evidence of the strengths of our method, which explores the benefits of leveraging multi-modal inputs in reinforcement learning tasks.

## A.4 Additional experiments on the DMControl Suite

we conduct additional experiments using the Mujoco-powered DeepMind Control Suite (DMControl) [4] with RGB and depth as input modalities on two challenging tasks (Cheetah, run and Walker, walk). The training and testing curves of HAVE and other comparable methods are given in A4. From the results, it is clear that HAVE also outperforms other methods in these robot control tasks.

## A.5 Ablation Study of Auxiliary Losses

We have carried out the corresponding experiments in the ClearNight environment of CARLA. The results are illustrated in Fig. A3 (c). Specifically, three terms are comprised: the global transition loss, the sum of individual transition losses, and the reward prediction loss. When compared to the

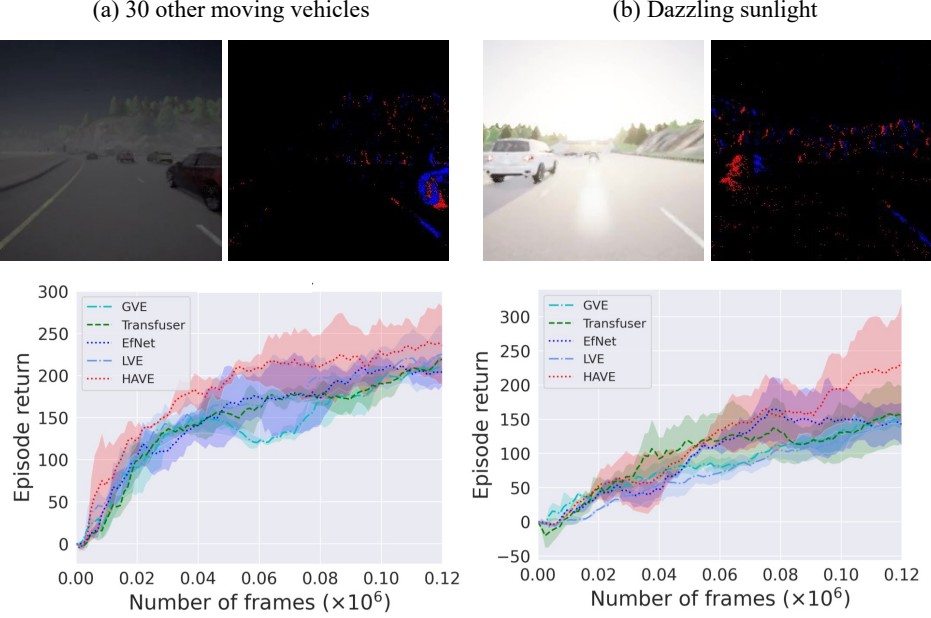

Figure A2: Visualization of challenging and extreme conditions and their corresponding experimental results.

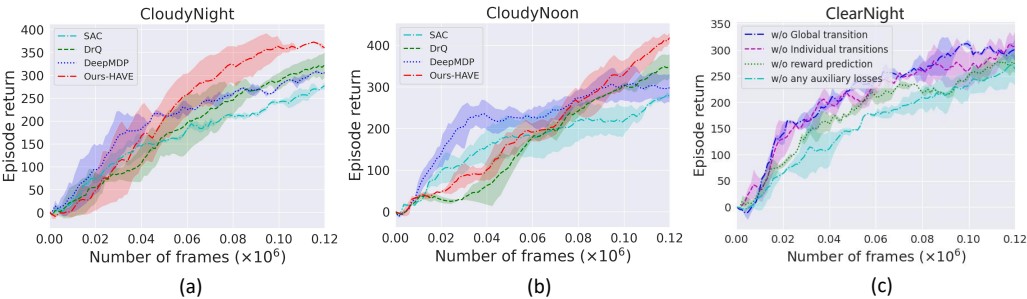

Figure A3: (a) and (b) are additional experimental results with three modalities on CloudyNight and CloudyNoon. (c) is auxiliary losses ablation.

full results of HAVE in Fig.4 (top left) of the main paper, Fig. A3 (c) reveals that omitting any of these terms negatively impacts model performance. Among them, the reward prediction loss is the most crucial, leading to the most significant performance drop. When all three terms are excluded (i.e., the 'w/o aux' curve in Fig. A3 (c)), there is a notable decline in performance.

## A.6 Ablation Study on the Design of Re-fusion

We need dynamic fusion over static ones (such as taking the mean of GVE and LVE values), since LVE does not always outperform when GVE provides accurate estimates, and conversely, GVE does not always excel when LVE estimates are precise. Their effectiveness is contingent upon the attributes of the current global modality, which can vary drastically depending on the environmental context. To address this variability, we employ a hypernetwork for a dynamic combination of both methods. While we have experimented with directly merging GVE and LVE, Table A1 indicates that, although direct fusion is feasible, the dynamic fusion through the hypernetwork proves more efficacious.

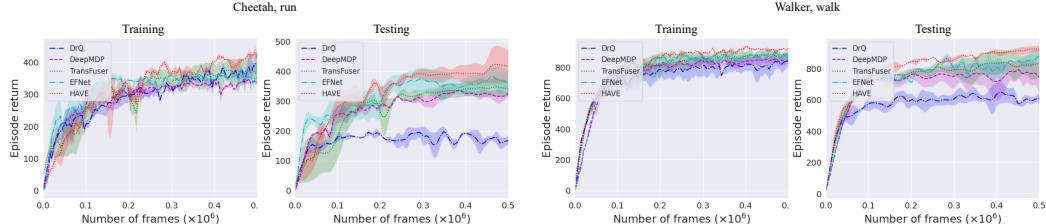

Figure A4: Performance of different algorithms on Mujoco-powered DeepMind Control Suite environments with RGB and depth frames as input modalities.

Table A1: Ablation study on the design of the re-fusion process on CARLA.

| | ClearNight | | ClearNoon | |
| Re-fusion method | Dynamic | Direct | Dynamic | Direct |
|---|---|---|---|---|
| ER | 319±71 | 307±67 | 336±76 | 307±84 |
| D(m) | 212±52 | 203±56 | 223±44 | 213±48 |

## B  Further Discussions

**Value-level vs. Feature-level Fusion** The value-level and feature-level dynamic fusion approaches in RL can be viewed as complementary paradigms: each have their unique strengths and contribute in different ways to the learning process. Feature-level dynamic fusion enables the combination of distinct feature sets originating from various modalities. In this way, the learning system can capitalize on the complementary and shared aspects of different modalities in a more nuanced manner. However, it may encounter difficulties when the alignment between modalities is not clear, or when modalities have conflicting information. Feature-level fusion can potentially amplify these issues, as it does not have the ability to weigh the importance of each modality based on the value it contributes to the task at hand. In contrast, value-based dynamic fusion, which operates on Q-values, allows for more informed decision-making as each modality can directly influence the final policy decision. As a result, it can effectively balance the different modalities based on their contribution to the expected reward, which can be particularly beneficial in environments where modalities offer divergent or contradictory information. In Fig. A6, we provide results on two more weather conditions. Taking together the results in Fig. A1, Fig. A2, and Fig. A6, it's apparent that the performance of one paradigm over another is dependent on specific environmental conditions. Nevertheless, our HAVE framework, which applies both fusion paradigms, consistently outperforms approaches using a single paradigm, demonstrating their complementarity.

**Bi-level Fusion vs. Uni-level Fusion** The hierarchical bi-level fusion approach in our multi-modal reinforcement learning framework allows a more refined balance between global and local modalities through its dual-weighting mechanism. By assigning individual weights $w^{M_i}$ to each local modality and subsequently modulating them with a task-related weight in the re-fusion network $\mathcal{H}$, the algorithm can adjust the contribution of each local modality twice: initially based on the intrinsic relevance of a modality, and subsequently, taking into account the overall task context. This

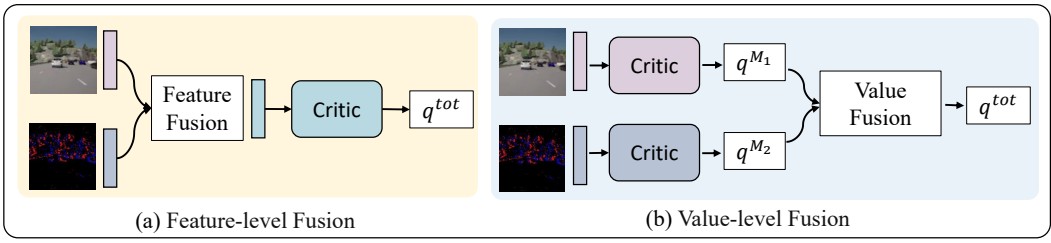

(a) Feature-level Fusion     (b) Value-level Fusion

Figure A5: Feature-level Fusion vs. Value-level Fusion.

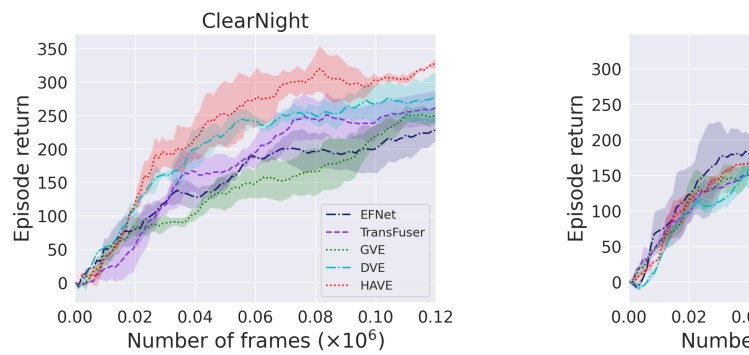

Figure A6: Comparison of HAVE, feature-level fusion methods (GVE, Transfuser, EFNet) and value-level fusion method (LVE).

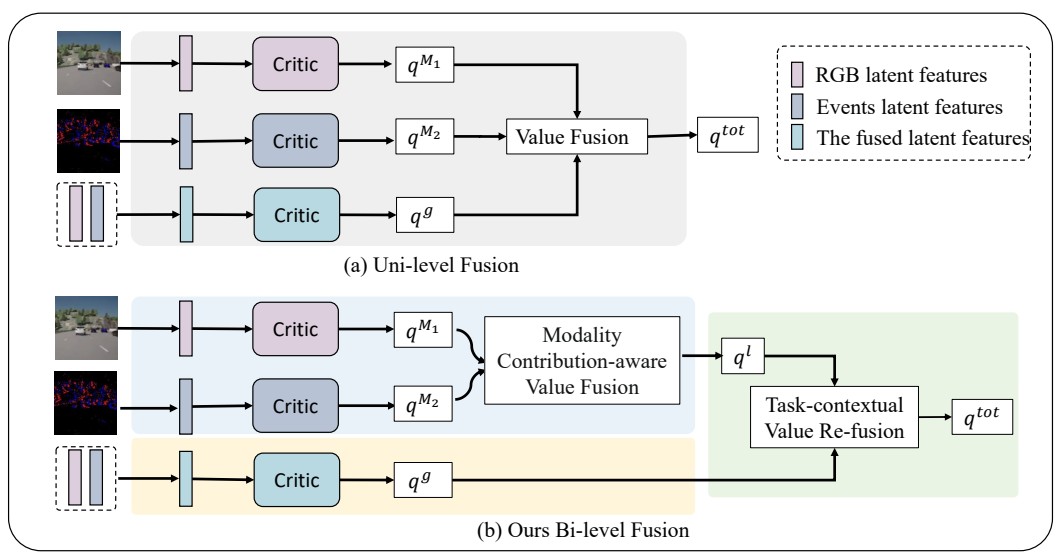

Figure A7: Hierarchical bi-level Fusion vs. uni-level Fusion.

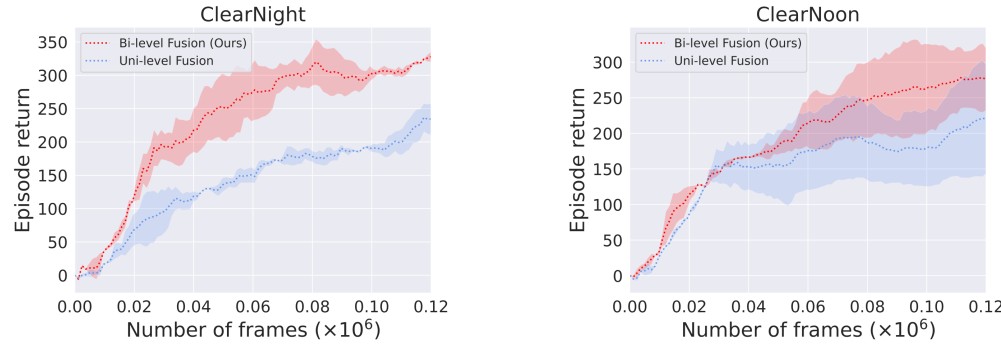

Figure A8: Performance comparison between hierarchical bi-level fusion and uni-level fusion.

double modulation facilitates a more nuanced integration of local and global modalities into the final value estimate. On the contrary, the uni-level fusion method shown in Fig. A7 affords only a single adjustment level per modality, which could potentially hinder the establishment of a precise balance between local and global modalities. Given that the global modality inherently comprises more information, it is plausible that its corresponding value weight might overshadow those of the individual local modalities, thereby downplaying their contributions. Our experimental results in Fig. A8 also demonstrate the advantage of hierarchical bi-level fusion over uni-level fusion.

**Relationship to Multi-agent RL** Multi-agent RL (MARL) focuses on the interaction between multiple learning agents, while multi-modal RL is about the fusion of information from multiple modality sources to enhance the learning of a single agent. Our adaptive value estimation method shares similar spirit of the value decomposition techniques in MARL [3, 5]. However, the methods and techniques used in the two areas can be quite different due to the distinct challenges each task faces.

In MARL, value decomposition involves splitting the joint action-value function into individual value functions for each agent. This is done to manage the complexity of the joint action space and to enable decentralized execution. The goal is to facilitate each agent's ability to take individual actions that contribute optimally to the collective reward, while considering the inter-dependencies of actions among the agents. This often involves a mixing network to aggregate the decomposed value functions, which allows for an intricate weighting system to balance the impacts of different agents' actions.

In contrast, our adaptive value estimation technique HAVE leverages information from different input modalities to predict a single action: The GVE uses a global critic to evaluate the global state, while the LVE involves separate critics for each modality. This approach aims to harness the unique insights each modality offers. The GVE and LVE are then combined to learn together to reduce the overall estimation error. Then the actor uses the updated value function from both GVE and LVE to improve its policy.

## C  Detailed Experimental Setup

In this section, we elaborate on the detailed experimental setup, including network architectures, environment setup of the CARLA benchmark, and hyper-parameters.

### C.1  Network Architecture

**Encoder** For each modality, we employ individual feature extractors to process the input data. The observed modalities consist of three stacked frames rendered at a resolution of $128 \times 128$. The input channel numbers of stacked RGB, events, and depth data are therefore 9, 15, and 3, respectively. All feature extractors share a similar network architecture comprising four convolutional layers, which generate $256 \times 6 \times 6$ feature maps at the last layer. After each convolutional layer, a ReLU layer is applied. Additionally, FC layers are placed after the channel-concatenated global modality feature map and each individual modality feature map, responsible for projecting the feature maps to 50-dimensional latent vectors $f^g$ and $\{f^{M_i}\}_{i=1}^d$.

**Actor and Critic Network** With the latent vectors $f^g$ and $\{f^{M_i}\}_{i=1}^d$ as inputs, the actor network generates parameters representing a Gaussian distribution over actions. This network architecture comprises three FC layers, with the first two FC layers followed by the non-linear activation function ReLU. On the other hand, the critic network shares a similar structure to the actor network, but it outputs a single value: the estimated Q-value for a given state-action pair. In our HAVE framework, we have $M_{d+1}$ critic networks in total. Among them, $d$ critic networks are dedicated to individual modality value estimation, evaluating the Q-value specific to each modality. Additionally, one critic network is specifically designed for global modality value estimation, providing an overall assessment across all modalities. By employing separate critic networks for each modality and a dedicated global modality critic network, we can effectively estimate the values associated with different modalities, enabling comprehensive value estimation in the joint learning of different modalities.

**Transition and Reward Prediction Network** In our framework, we employ a total of $M_{d+1}$ transition networks, including individual transition networks for each modality and a transition network designed for the global modality. Here we take the global modality transition network as an ex-

ample. Following the experimental setup described in [1], the transition network takes the current latent vector observation $f_t^g$ of the global modality and the corresponding action $a_t$ as inputs. It then generates the inferred latent vector for the next global modality observation using two linear layers. These linear layers have a hidden dimension of 512, and the first layer is followed by a ReLU activation function. The purpose of the transition network is to model the dynamics of the global modality and predict the next latent vector observation based on the current observation and the chosen action. Similar to the global modality, the transition networks for other local modalities follow a similar architecture but have different input and output dimensions based on the specific modality. Each individual modality transition network is responsible for predicting the next latent vector observation for its respective modality. Furthermore, there is a single reward prediction network in our framework. This network shares a similar architecture with the transition network but outputs a predicted reward with a dimension of one. The reward prediction network aims to estimate the expected reward associated with a given state-action pair, providing crucial feedback for reinforcement learning.

## C.2  Environment Details

### C.2.1  Weather Conditions

The eight weather conditions (ClearNight, ClearNoon, CloudyNight, CloudyNoon, HardRainNight, HardRainNoon, WetNight and WetNoon) used in our study are predefined based on the default parameters of the CARLA simulation environment. In addition to the eight predefined weather conditions, we consider specific weather hyper-parameters of some extreme weathers, which are outlined in detail in Table A3. These hyper-parameters provide a comprehensive understanding of the weather conditions simulated during our study.

### C.2.2  Modalities

**RGB Camera**: The "RGB" camera in our setup functions as a conventional camera, capturing rich texture and color information of the scene. However, traditional cameras face two prominent challenges. The first issue is motion blur, which occurs when the speed of motion in the scene surpasses the camera's sampling rate. This results in blurry images. Although motion blur can be mitigated through algorithms, it often incurs significant computational costs, making it unsuitable for real-time applications. The second challenge is limited dynamic range, which refers to issues of underexposure or overexposure caused by varying lighting conditions. In instances of intense sunlight, traditional cameras may struggle to capture objects within the field of view. Consequently, we provide key parameters related to motion blur and dynamic range settings to address these challenges effectively, as shown in Table A2.

**Events Camera**: Event cameras, such as the Dynamic Vision Sensor (DVS), overcome the limitations of traditional cameras at the sensor level. Unlike conventional cameras, event cameras solely focus on observing "motion" or, more precisely, "changes in brightness" within the scene. When a brightness change occurs, the event camera outputs a binary signal (1 or -1) indicating the brightness change of the corresponding pixel. Event cameras exhibit distinct advantages, including a significantly wider dynamic range (140 dB versus 60 dB) than traditional cameras, absence of motion blur, and high temporal resolution (in the order of microseconds). *These characteristics make event cameras highly suitable for vision-based reinforcement learning tasks, as they excel at capturing dynamic information.* However, it is important to note that event cameras do not generate an image if there is no pixel difference between two consecutive synchronous frames. Similar to conventional cameras, event cameras may also experience hot spot noise, which can arise from pixel damage or issues with the charging and discharging mechanism, resulting in the generation of high-frequency events. To replicate the image characteristics produced by actual event cameras, we introduce noise into our simulation. These noises emulate the inherent characteristics of event cameras. The specific parameter settings for the noise generation are summarized in Table A2.

**Depth Camera**: The depth camera is not the main focus of our research, but it serves to validate the effectiveness of our proposed method under multiple modalities. Therefore, no specific treatment or customization is applied to the depth camera. We follow the default settings provided by the CARLA simulation platform to obtain depth information. While it is not a primary focus, the depth camera's default configuration allows us to incorporate depth information into our experiments and evaluate the performance of our method.

Table A2: Hyper-parameter setting used in our experiments. The names of these parameters can be found in the documentation of CARLA.

| RGB | |
|---|---|
| Exposure_speed_up | 3.0 |
| Exposure_speed_down | 1.0 |
| Blur_amount | 1.0 |
| Motion_blur_intensity | 1.0 |
| Motion_blur_max_distortion | 0.8 |
| Motion_blur_min_object_screen_size | 0.4 |
| Lens_flare_intensity | 0.2 |
| Shutter_speed | 100.0 |
| **Events** | |
| Positive_threshold | 0.3 |
| Negative_threshold | 0.3 |
| Sigma_positive_threshold | 0.05 |
| Sigma_negative_threshold | 0.05 |
| **Other setting of environment** | |
| Num_cars | 20 |
| Num_cameras | 1 |

### C.2.3 Reward Setting

For autonomous driving in CARLA, the goal of agent is to drive as far as possible on highway without collision under diverse weather conditions. Thus the reward function is designed similar to [6]:

$$r_t = v_{ego}^\top \hat{u}_{highway} \cdot \Delta t - \lambda_c \cdot collision - \lambda_s \cdot |steer| - \lambda_b \cdot brake \tag{1}$$

where $v_{ego}^\top$ denotes the velocity vector of our agent vehicle, $\hat{u}_{highway}$ is the unit vector of highway, and $\Delta t = 0.1$ is the discretized the simulation time. $\lambda_c$, $\lambda_s$ and $\lambda_b$ are set to 0.001, 0.1 and 0.1 respectively. The first term is designed to encourage the vehicle to travel as far as possible along the highway. The last three terms are put in place to ensure the vehicle avoids collisions, minimizes excessive steering, and refrains from abrupt braking.

### C.3 RL-related hyper-parameters

The RL-related hyper-parameters of our experiments are detailed in Table A4. For a fair comparison, we adopt the same hyper-parameters as in [2].

Table A3: Weather hyper-parameters used in our experiments.

| Weather Parameters | Dazzling Sunlight |
|---|---|
| Cloudiness | 0.0 |
| Precipitation | 0.0 |
| Precipitation_deposits | 0.0 |
| wind_intensity | 0.0 |
| fog_density | 0.0 |
| fog_distance | 1000.0 |
| wetness | 0.0 |
| sun_azimuth_angle | 270.0 |
| sun_altitude_angle | 10.0 |

### C.3.1 Hardware Details

**Computing infrastructure** We train all models with a server equipped with NVIDIA GeForce RTX 3090 GPUs and a 64-core AMD EPYC 7H12 2.6GHz CPU Processor.

Table A4: RL-related hyper-parameters used in our experiments.

| Hyperparameter | Value |
|---|---|
| Frame rendering | $128 \times 128$ |
| Stacked frames | 3 |
| Action repeat | 1 |
| Batch size | 128 |
| Discount factor $\lambda$ | 0.99 |
| Init steps | 1,000 |
| Episode length | 1,000 |
| Learning algorithm | Soft Actor-Critic (SAC) |
| Number of frames | 120,000 |
| Replay buffer size | 100,000 |
| Optimizer (encoder, actor, critic) | Adam ($\beta_1 = 0.9$, $\beta_2 = 0.999$) |
| Optimizer (transition and reward prediction network) | Adam |
| Learning rate (encoder, actor, critic) | 1e-3 |
| Learning rate (transition and reward prediction network) | 1e-3 |
| Learning rate ($\alpha$ in SAC) | 1e-3 |
| Batch size | 128 |
| Transition and reward prediction network update frequecy | 1 |
| Actor update freq update frequency | 1 |
| Critic target update freq update frequecy | 2 |

## D   Potential Negative Societal Impacts

Autonomous systems powered by reinforcement learning could be vulnerable to adversarial attacks, leading to malfunctioning or rogue behavior. In the worst case, such systems could pose physical dangers if they operate machinery or vehicles. On the other hand, reinforcement learning models, especially those dealing with multi-modal data, can be complex and hard to interpret. This lack of transparency can lead to a lack of trust in their decisions, especially in critical areas like healthcare or autonomous vehicles.