# OpenReview forum: "Hierarchical Adaptive Value Estimation for Multi-modal Visual Reinforcement Learning"
_NeurIPS.cc/2023/Conference — NeurIPS 2023 poster_

### Official Review · Reviewer_HRjH · 2023-06-17

**Soundness:** 2 fair
**Presentation:** 3 good
**Contribution:** 2 fair
**Rating:** 3
**Confidence:** 4

**Summary:**

This paper presents a new vision-based reinforcement learning algorithm for autonomous driving scenarios. In previous feature fusion methods, a single critic could amplify dominant features and obscure other features. To address this issue, the authors propose a local critic for each feature and combine them into a global critic using attention fusion. Additionally, the authors introduce a task-contextual re-fusion component to rebalance the estimator. The algorithm achieves promising performance on the CARLA driving task.

**Strengths:**

1. The writing in the paper is easy to follow, and the figures are clear and understandable.
2. The proposed algorithm has a simple and effective structure. The improvements made to existing methods in the framework section are intuitive.
3. The argumentation in the "Further Analyses" section of the paper is thorough and provides a good explanation for the potential sources of performance improvement brought by the algorithm.

**Weaknesses:**

1. The decomposition of the Q-network lacks novelty. Although this paper focuses on vision-based RL, the approach of using a separate critic for each modality and obtaining a global critic using self-attention is similar to some existing methods in the field of multi-agent RL[1]. The paper does not adequately highlight the differences in information processing for vision-based tasks compared to other types of input.

2. The central contribution of the paper lies in the proposed framework. However, apart from the front-end modality fusion and task-specific re-balance, all the content mentioned in section 3.5 appears to be supplementary to existing research[2][3] rather than original improvements. Therefore, I believe there is a lack of substantial contribution in the paper.

ref
[1] QMIX: Monotonic Value Function Factorisation for Deep Multi-Agent Reinforcement Learning
[2] Soft actor-critic: Off-policy maximum entropy deep reinforcement learning with a stochastic actor.
[3] Deepmdp: Learning continuous latent space models for representation learning

**Questions:**

In Section 4.2, I noticed that the authors mentioned, "Since TransFuser and EFNet only support two modalities, we directly compare our method with SAC, DrQ, and DeepMDP." This suggests that these two baseline algorithms are not suitable for the task environment used in the current paper. Then why use them as strong baselines to compare?

**Limitations:**

The experimental setup is not sufficiently fair. Although the baselines used for comparison include newer multimodal vision algorithms, they were originally designed for traditional visual tasks. In the tasks used for algorithm validation in this paper, there is a lack of a strong baseline that specifically demonstrates the superiority of the proposed algorithm.

---

> ### Author Rebuttal · Authors · 2023-08-10
>
> We would like to express our sincere gratitude for your thoughtful review of our paper. Your positive remarks on our writing and analysis are truly appreciated, and we feel encouraged by your feedback. Regarding your concerns about the novelty of our work, we sincerely apologize if our presentation did not sufficiently highlight our unique contributions. We value your feedback and hope the following responses can clarify and address any misunderstandings that might have arisen.
>
> > 1. lacks novelty compared with QMIX.
>
> Thank you for your comments. Your observation regarding the similarity between HAVE and multi-agent is incisive. We kindly ask for your patience and attention as we delve into a more detailed explanation below.
>
> Specifically, HAVE intends to answer: “What is the appropriate paradigm to determine state-action values considering the modality-specific contribution?” Through harnessing resources of the RL field itself, we observed a similarity between the roles of “agent'' in multi-agent RL and “modality'' in multi-modal RL: both exhibit distinctive contributions to the final reward and require effective cooperation to optimize the global decision. This resemblance highlights a shared principle across different domains within RL, which by itself is a new kind of discovery not mentioned by existing works.
>
> However, the devil is in the details. The subtle yet critical aspects, when examined closely, differentiate our method from the multi-agent ones. Particularly,  our method aims to solve totally different problem compared with QMIX :
>
>
> 1. QMIX is proposed for multi-agent cooperative learning tasks where multiple agents share a global or collective reward. QMIX aims to decompose the global Q value for each agent's action, which is further used for training each agent's policy.
>
> 2. The proposed HAVE aims to solve “How to predict Q value of a unified action from a single agent considering the modality-specific contribution under inconstant environment dynamics?” Different with the general multi-modality supervised learning method, the Q value doesn't have true groundtruth during learning, and the current learning goal of Q is related to the current critic. Hence, the feature-level fusion is inadequate, and it is necessary to study how to estimate more accurate Q value by fusing multi-modality estimators. Therefore we have hierarchical and dynamic value fusion. In contrast, QMIX does not consider either the choice of fusion paradigm or bi-level fusion of values.
>
>
>
> > 2. ...all the content mentioned in section 3.5 appears to be supplementary to existing research (SAC[2] and DeepMDP [3]) rather than original improvements.
>
> Our method focuses on multi-modality reinforcement learning on visual control tasks. SAC and DeepMDP are two well-known and classical baselines on single modality reinforcement learning (RL) and visual presentation learning of RL. It is reasonable and general to study our problem based on them. In fact, there are many approaches that take them as the baseline, such as:
>
> [1] Image augmentation is all you need: Regularizing deep reinforcement learning from pixels. ICLR. 2020.
> [2] Curl: Contrastive unsupervised representations for reinforcement learning. ICML.2020.
> [3] Stabilizing deep q-learning with convents and vision transformers under data augmentation. NeurIPS. 2021.
> [4] Learning invariant representations for reinforcement learning without reconstruction. ICLR. 2021.
> [5] Masked contrastive representation learning for reinforcement learning. TPAMI 2022.
> [6] Information Optimization and Transferable State Abstractions in Deep Reinforcement Learning. TPAMI 2022.
> [7] Dreaming: Model-based reinforcement learning by latent imagination without reconstruction. ICRA. 2021.
> [8] Isolating and Leveraging Controllable and Noncontrollable Visual Dynamics in World Models. NeurIPS. 2022.
> [9] Denoised mdps: Learning world models better than the world itself. ICML. 2022.
> [10] Reinforcement learning with action-free pre-training from videos. ICML, 2022.
> [11] Masked world models for visual control. CoRL, 2023.
>
> SAC is used as the baseline in [1,2,3,4,5,6], and DeepMDP is used in [7,8,9,10,11].
>
>
> > 3.  Why use TransFuser and EFNet as strong baselines to compare?
>
> We apologize for the confusion caused by our previous statement in Sec. 4.2. Our statement was primarily grounded in concerns about computational complexity and practical implementation. Although it's technically possible to scale both methods to accommodate more than two modalities, such an adaptation would require significant modifications to the implementation, along with a quadratic increase in computational complexity due to cross attention mechanism.
>
> In addition, most of the experiments in our paper adopted two modalities. Since EFNet and TransFuser are proven very robust with paired modalities, we use them as strong baselines for our comparison following [1-3] and only omit them in our three-modality tests, which is only a small portion of the total experiments.
>
> [1] Safety-enhanced autonomous driving using interpretable sensor fusion transformer. PMLR 2023
>
> [2] Plant: Explainable planning transformers via object-level representations. CoRL 2022
>
> [3] Rgb-event fusion for moving object detection in autonomous driving. ICRA 2023
>
> >4. The experimental setup is not sufficiently fair.
>
> Thanks for your suggestion. We introduced a new baseline named MuMMI [1], which is also designed to manage multiple sensors in reinforcement learning scenarios. The results are detailed in Table 3 of the rebuttal PDF. Relative to other methods, HAVE has a pronounced advantage in performance.
>
> [1] Multi-modal mutual information (MuMMI) training for robust self-supervised deep reinforcement learning. ICRA. 2021.

---

> > ### Author Response · Authors · 2023-08-19
> > **Additional information on novelty and contribution of HAVE**
> >
> > Considering you might have further concerns regarding the novelty and contribution of our work, we offer additional information here. This comment focuses on **detailed technical differences**, as well as the **inspirations, thoughts, and rationales** behind HAVE.
> >
> > We first give a brief comparison of multi-agent RL (MARL) methods (e.g., QMIX) and HAVE in the following table:
> >
> > | Method | QMIX | HAVE |
> > |:---|:---|:---|
> > | **Problem Focus** | Addresses decentralization of actions with a global reward during training. | Leveraging information from different sensors/input sources to take a single action. |
> > | **Challenges** | Assigning collective Q values to each agent having individual observations and actions. | Modeling a single accurate Q value from multi-modal perceptual data for a united action. |
> >
> >
> > At a high level, these will serve as the guidelines for our individual technical design. Going deeper into the details, we believe the following points call for emphasis:
> >
> > &nbsp;
> >
> > ### **1. Varied technical nuances and lines of reasoning**
> >
> > As you may have discovered, QMIX uses a $\textcolor{red}{\text{monotonic hypernet}}$ to maintain monotonicity between individual agent’s Q values and global Q value, adhering to the Individual-Global-Max (IGM) property [1] (i.e., $\frac{\partial Q_{tot}}{\partial {Q_{a}}} \geq 0, \forall {a} \in A$) in MARL. IGM ensures decentralized policies align with centralized ones during local greedy actions. However, **it is not applicable to multi-modal RL** because 1) modalities collaborate to decide a final action, rather than determining actions greedily, and 2) Estimated Q values may be affected by gaining/losing vital information due to modality characteristics and sensor noises, which breaks monotonicity. For instance, while a red traffic light may not be captured by event sensors and results in a high estimated Q value on thrust, the global Q value should decline in such cases. Therefore, although obtaining a separate critic for each modality is a first step choice, how to make the best of them remains unknown due to the inherent difference between MARL and our task.
> >
> > To crack this, interactions between modalities and their relationships with the global modality are crucial. However, $\textcolor{red}{\text{QMIX's hypernets do not explicitly model these local-global interactions}}$. Furthermore, we also aim to avoid modality dominance as stated in Remark 1 in Sec.3.6 of the paper. To achieve these, the LVE paradigm in HAVE adopts $\textcolor{green}{\text{cross-attention between Q values and the local/global modality features}}$ with better results. While the attention mechanism still enforces IGM due to Softmax's non-negative weights, HAVE's $\textcolor{green}{\text{task-level re-fusion}}$ employs $\textcolor{green}{\text{non-monotonic hypernets }}$ to re-balance the values from LVE and GVE based on environmental rewards. We do not enforce positive weights from the hypernets, thereby liberated from the IGM constraint between the final fused value and individual modality values. Such a bi-level process **jointly archives 1) the prevention modality dominance and 2) the discovery of opposing trends between individual Q values and global Q value**, which cannot be achieved by a single step of value decomposition and merging.
> >
> > To our knowledge, no existing MAML method adopts such a bi-level fusion with the IGM constraint lifted. Meanwhile, current multi-modal RL methods also overlook modality-specific values. Therefore, we believe we have contributed an efficient multi-modal RL framework by re-thinking modality relationships with a novel learning paradigm.
> >
> > It should also be noted that attention and hypernets are both classical techniques widely adopted by researchers, while the motivation, reasoning and implementation details can vary greatly for each method.
> >
> > &nbsp;
> >
> > ### **2. MARL methods will NOT work well directly on multi-modal RL (+Evidence)**
> >
> > As we analyzed above, the MARL methods with IGM constrain may not apply to our task. During the exploration of our work, we tested QMIX directly on multi-modal RL. However, the results were quite unsatisfactory (See the performance table below on CARLA). The results demonstrated that solely decomposing critics for each modality alone is not enough for multi-modal RL. We release these results to clarify that HAVE is a standalone method with thoughts and trials and errors behind it, which is not simply a technical re-use of MARL methods.
> >
> > | Scenario  | QMIX    | Ours-LVE | Ours-HAVE |
> > |-----------|---------|----------|-----------|
> > | Clearnight| 231±85  | 274±68   | 319±71    |
> > | Clearnoon | 264±78  | 294±82   | 336±76    |
> >
> > Reference:
> >
> > [1] QTRAN: Learning to Factorize with Transformation for Cooperative Multi-Agent Reinforcement Learning, ICML 2019
> >
> > &nbsp;
> >
> > Please kindly let us know if our responses have addressed your concerns, or if any further clarification is desired.

---

> > > ### Author Response · Authors · 2023-08-19
> > > **Additional information on novelty and contribution of HAVE (continued)**
> > >
> > > To provide a clearer understanding, we offer a diagrammatic comparison between HAVE and QMIX below:
> > >
> > > HAVE
> > > ---
> > > $Modality_1$↘
> > > $Modality_2$ →Actor→$\textcolor{red}{\text{action}}$
> > > $Modality_n$↗
> > >
> > > $Modality_1$,$\textcolor{red}{\text{action}}$   →$Critic_1$→$Q_{1}$↘
> > > $Modality_2$,$\textcolor{red}{\text{action}}$   →$Critic_2$→$Q_{2}$→$Q_l$
> > > $Modality_n$,$\textcolor{red}{\text{action}}$   →$Critic_n$→$Q_{n}$↗  &emsp;&emsp; ↘
> > > &emsp;&emsp;&emsp;&emsp;&emsp;&emsp;&emsp;&emsp;&emsp;&emsp;&emsp;&emsp;&emsp;&emsp;&emsp;&emsp;&emsp;&emsp;&emsp;&emsp;&emsp; $Q_{fusion}$  →Loss
> > > $Modality_{1,2,...,n}$,$\textcolor{red}{\text{action}}$   →$Critic_{n+1}$→&nbsp;&nbsp;$Q_{g}$↗
> > >
> > > QMIX
> > > ---
> > > $Agent_1$→$Qnet_1$→$Q_{11},Q_{12},...$→$\textcolor{red}{\text{$action_2$}}$ choosen for $agent_1$→$Q_{12}$↘
> > > $Agent_2$→$Qnet_2$→$Q_{21},Q_{22},...$→$\textcolor{red}{\text{$action_1$}}$ choosen for $agent_2$→$Q_{21}$→ $Q_{tot}$→Loss
> > > $Agent_n$→$Qnet_n$→$Q_{n1},Q_{n2},...$→$\textcolor{red}{\text{$action_j$}}$ choosen for $agent_n$→$Q_{nj}$↗
> > >
> > > In conjunction with the provided diagram and the prior statements, it should be clear that HAVE and QMIX serve as distinct frameworks, catering to varied tasks and objectives.
> > >
> > > Within the comprehensive framework, the fusion of multi-modal Q-values bears resemblance to the multi-agent fusion approach in QMIX. Nonetheless, even when examining the operations of this specific step, our approaches vary in multiple aspects:
> > > (1) Weighting: While QMIX adheres to the IGM principle and ensures positivity, our modalities don't encounter this issue. In our case, a modality's prediction can occasionally be erroneous or negative, so we don't impose such constraints.
> > > (2) Fusion Mechanism: In our approach using a cascading technique, the fusion of Q-values predicted among individual modalities and the fusion of Q-values estimated between single modality and fused modality features do not operate at the same hierarchical tier (for a detailed explanation, refer to section 3.4 of the main paper and the second paragraph of section B in the supplementary materials
> > > ). In contrast, QMIX operates without the need for cascading and all agents participate in fusion at a uniform level.  In the performance table of our last comment, we replaced our fusion approach with the QMIX fusion method for the experiment, the inferior performance of QMIX-style fusion further highlights the advantages of the cascading approach of HAVE.

---

> > > > ### Author Response · Authors · 2023-08-21
> > > > **Other issues the reviewer may be concerned about**
> > > >
> > > > Dear Reviewer HRjH,
> > > >
> > > > We have taken note of the absence of further feedback following our previous clarifications. We interpret this as potential lingering reservations or unsatisfactory responses on our end. While we have tried our best to address your concerns surrounding the contribution of our work in previous comments, we recognize that there might still be other aspects of our research that you have reservations about. With that in mind, we would like to offer further clarification on the other points you have raised to ensure a comprehensive understanding of our methodology and contributions.
> > > >
> > > > Regarding **supplementary to existing research (SAC/DeepMDP)**, we want to reiterate that we intend not to develop a base RL algorithm but rather a comprehensive value estimation framework tailored for multi-modal RL. As we state in the rebuttal post, both SAC (maximizes the expected reward plus the entropy of the policy) and DeepMDP (predicts the rewards and the distribution over
> > > > next latent states) are classical base RL algorithms that numerous works built upon. We put their description in the method section considering narrative coherence, which serves as a technical starting point for our method. Instead, we value and intend to contribute the “front-end modality fusion and task-specific re-balance” you mentioned but set aside.
> > > >
> > > > Regarding the **experimental setup**, there are two main reasons why we chose TransFuser and EFNet baselines. First, given that the core objective of visual reinforcement learning is to make decisions directly from pixel data, conventional visual methodologies naturally serve as vital benchmarks for this domain. An example of this is DrQ, which emphasizes regularization in RL from pixels via data augmentation. Second, to the best of our knowledge, there is a noticeable dearth of research in the domain of multimodal vision-based reinforcement learning, leading us to compare it with a limited number of studies. In this context, we introduce TransFuser and EFNet, both stemming from traditional visual paradigms, as part of our comparative baselines. Simultaneously, following a more comprehensive investigation, we integrated a new baseline, MuMMI, which aims to learn a shared representation between modalities in model-based RL. The outcomes of this comparison are presented in Table 3 of the rebuttal PDF. When benchmarked against MuMMI, HAVE distinctly outperforms in terms of episode reward.
> > > >
> > > > We thank your effort in reading and considering all our responses. The deadline for our discussion is approaching. If you have any remaining concerns, please don’t hesitate to discuss them with us. We are more than willing to address any questions you may have.

---

> ### Comment · Senior_Area_Chairs · 2023-08-21
> **final discussions**
>
> Dear Reviewer,
>
> As discussions come to an end soon, this is a polite reminder to engage with the authors in discussion.
> Please note we take note of unresponsive reviewers.
>
> Best regards,
> \
> SAC

---

### Official Review · Reviewer_P2RM · 2023-07-03

**Soundness:** 3 good
**Presentation:** 3 good
**Contribution:** 3 good
**Rating:** 7
**Confidence:** 4

**Summary:**

The authors propose a Local modality-customized Value Estimation (LVE) paradigm, which dynamically estimates and adjusts the importance weight of each modality from the perspective of value-level to compensate for the multi-modal vision-based RL methods that usually use fused modality features for learning Global Value Estimation (GVE), which may be insufficient in policy learning. Furthermore, the authors propose a task-contextual re-fusion procedure to achieve a task-level rebalancing of estimates from feature and value levels. Experimental CARLA benchmark results show the improvement of multi-modal vision-based autonomous driving RL tasks.

**Strengths:**

- The originality and novelty of the paper are solid. Inspired by some ideas in algorithms in Multi-Agent RL, an ingenious weighted local value estimation method suitable for multi-modal RL is proposed. And a task contextual re-fusion process is designed to combine better global value estimation and weighted local value estimation to obtain better multi-modal value estimation.
- The paper is written with high quality. The writing structure is clear and organized, the sentences are fluent and smooth without grammatical errors, and the language style is simple and easy to understand. Although there is no rigorous theoretical analysis, it gives some interesting insights from mathematics. In addition, the code and appendices are well documented and provide useful analysis and explanation.
- The significance of the paper is fine. Developing better algorithms to solve multi-modal RL tasks is an important research topic.

**Weaknesses:**

- The experiments for this paper are still not adequate. The entire manuscript has always only conducted experiments on an autonomous driving environment CARLA, and it is difficult to evaluate whether HAVE is suitable for other broader multi-modal RL environments (such as simulated robot control). In addition, this work only conducts experiments on tasks of multiple visual modalities and does not consider more common modal inputs such as text and voice.
- The paper lacks further theoretical analysis. Although two interesting remarks are in Section 3.6 of the manuscript, the theoretical insight obtained is still relatively shallow. Lack of more in-depth theoretical analyses, such as the overall convergence of the algorithm and whether the algorithm optimizes some lower bounds under the multi-modal RL setting.

**Questions:**

- The baselines compared in this paper are still not representative and pertinent. This paper mainly focuses on the multi-modal RL task of visual input, but there is no comparison with the classic Vision-based RL work, and there are more suitable baselines than SAC, DrQ, and DeepMDP. For example:
  - Visual Reinforcement Learning with Imagined Goals, NeurIPS 2018
  - Improving Sample Efficiency in Model-Free Reinforcement Learning from Images, AAAI 2021
  - Stabilizing Deep Q-Learning with ConvNets and Vision Transformers under Data Augmentation, NeurIPS 2021
- Observing Figure 3 and Figure 4, it seems that the training curves have not yet converged. If the training process is extended, would HAVE be surpassed by other comparative methods?
- Why do TransFuser and EFNet only support two modalities, as mentioned in article 4.2?
- Would using neuromorphic event signals lead to totally fair comparisons? The algorithm's performance may be related to the peculiarity of the modality itself. For example, TransFuser may have its advantages when using the Bird’s Eye View (BEV) modality, but it may be challenging to have a good command of neuromorphic event signals
- Different local areas of each modality could also have different importance. Can HAVE balance the value of this finer-grained "modality"? And how?

**Limitations:**

The authors explained the limitations of their work well. For example, this work only conducts experiments on RL tasks of multiple visual modalities and does not consider more extensive multimodal input such as text and voice. In addition, the authors explained their potential negative social impact in the appendix. I do not see an obvious negative societal impact.

---

> ### Author Rebuttal · Authors · 2023-08-10
>
> We are truly grateful for your time and effort invested in reviewing our work. Your positive feedback has been both encouraging and instrumental for the further improvement of our paper.  Herein, we address the points raised in your comments:
>
> >1. The experiments for this paper are still not adequate due to only with CARLA.
>
> Thanks for highlighting this. Following your advice, we have conducted experiments on an additional environment (DeepMind Control Suite for robot control). Please refer to our general response posted above for detailed results. Experiment results show that HAVE also works effectively in the new environment.
>
> >2. The paper lacks further theoretical analysis.
>
> Thank you for pointing out the need for a deeper theoretical analysis. We recognize the importance of a comprehensive theoretical foundation. The remarks in Section 3.6 were our initial efforts to provide theoretical insights into our approach. However, despite our earnest efforts to provide more theoretical proof, we encountered challenges that prevented us from establishing definitive results at this stage given the constraints of our current rebuttal timeline.
>
> We believe the empirical results presented do offer a validation of our method's effectiveness in the multi-modal RL setting. We sincerely value your feedback and, in future iterations of this work, aim to delve deeper into the theoretical aspects you've highlighted. Incorporating such a comprehensive analysis would undoubtedly strengthen our work, and we're committed to pursuing this in subsequent research.
>
> >3. The baselines compared in this paper are still not representative and pertinent.
>
> Thanks for the valuable suggestion. In Table 3 of our uploaded rebuttal PDF, we give additional results of three baselines. Two of which are based on your suggestion (SAC+AE [1] and SVEA [2]) and the rest (MuMMI[3]) is a multi-modal vision-based RL method. Compare with Table 1 in the main paper, we see that HAVE still possesses advantages over these baselines.
>
> >4. Observing Figure 3 and Figure 4, it seems that the training curves have not yet converged.
>
> Thanks for this astute observation. We conducted further experiments focusing on the figures that seemed to have the most serious convergence issues, namely Fig. 3 (left) and Fig. 4 (bottom left). For each figure, we evaluated two methods that appeared to be the least converged. The findings, presented in Fig.2 (b) and Fig.2 (c) of the rebuttal PDF, indicate convergence by 150K steps, with HAVE consistently outperforming the others. In subsequent revisions, we plan to present results up to 150K steps for all experiments.
>
> >5. Why do TransFuser and EFNet only support two modalities?
>
> We apologize for the confusion caused by our previous statement in Sec. 4.2. Our statement was primarily grounded in concerns about computational complexity and practical implementation. For clarity, the cross-modality attention mechanisms embedded within EFNet are designed for paired modalities. In this design, one modality calculates the query tensor, while the other calculates the key and value tensors. We also observed that only two modalities are utilized in both the Transfuser paper and its official implementation. Although it's technically possible to scale both methods to accommodate more than two modalities, such an adaptation would require significant modifications to the implementation, along with a quadratic increase in computational complexity. For instance, with EFNet, a cross-modal attention operation must be performed for every modality pair. In comparison, the complexity added by extra modalities only scales linearly with HAVE. We will revise our statement to articulate this more precisely and prevent further misunderstandings.
>
> >6. Would using neuromorphic event signals lead to totally fair comparisons?
>
> We fully understand your concern. Indeed, it can generally be difficult to disentangle the contribution of an architecture’s modality compatibility from the effectiveness of its multi-modal learning mechanism. To maintain as fair a comparison as possible, for TransFuser and EFNet, we utilized the same observation encoder as with HAVE. With EFNet, we can safely assume there is no compatibility issue, as it is specifically designed for event signals and has proven to be highly effective. TransFuser was originally designed for LiDAR BEV and RGB. However, the architectures of TransFuser and EFNet share considerable similarities (both are transformers with cross-modality attention), and LiDAR BEV has characteristics similar to event frames (e.g., sparsity). Based on these observations, we conjecture that the influence of compatibility might be negligible. In our experiments, TransFuser outperforms EFNet in certain environments, which supports our conjecture. For a more comprehensive comparison of the modality generalization ability of HAVE and TransFuser, we also tested them in DeepMind Control Suite under robot control environments, where the inputs are RGB and depth. Experimental results show that HAVE maintains its advantage over TransFuser.
>
> >7.  Can HAVE balance the value of this finer-grained "modality"?
>
> Yes, it can. Specifically, the LVE in HAVE works by maintaining separate critic networks for each modality, allowing it to estimate individual state-values. Each critic in HAVE can therefore assign importance to local areas within its modality by activating only significant regions in the feature maps. As an illustration, we visualize the heatmap of different critic networks in Fig.2(f) in the rebuttal PDF, which aligns with our analysis.
>
>
> **Reference**
>
> [1] Improving Sample Efficiency in Model-Free Reinforcement Learning from Images, AAAI 2021
>
> [2] Stabilizing Deep Q-Learning with ConvNets and Vision Transformers under Data Augmentation, NeurIPS 2021
>
> [3] Multi-modal mutual information (MuMMI) training for robust self-supervised deep reinforcement learning." ICRA. 2021

---

### Official Review · Reviewer_9NFJ · 2023-07-06

**Soundness:** 2 fair
**Presentation:** 2 fair
**Contribution:** 2 fair
**Rating:** 5
**Confidence:** 4

**Summary:**

This paper proposes a fusion scheme for multi-modal RL, with a particular focus on fusing the modalities in estimating the value function. The authors propose a combination of global value estimation, local value estimation and another conditioned fusion of the global and local value estimation. The approach is evaluated in particular on fusing RGB and event-based data in an autonomous driving scenario with the CARLA simulator.

**Strengths:**

The paper proposes a novel scheme of "value level fusion" that is new to me.

**Weaknesses:**

- The authors propose a very complex scheme for fusing the q-value estimate based on the different modalities, having GVE, LVE and then another fusing of those two with a hyper network. Especially since the policy is selecting actions based on a simple concatenation of feature vectors of both modalities.

- There is a very high spread on the results which makes me wonder how significant the improvement is of the HAVE approach. Also, from Fig 3 and 4, it seems that the other methods are not necessarily converged yet. It would be interesting to see how the curves evolve if training would continue.

- All experiments are done on this single environment with a custom reward function. It would be more convincing if the approach was demonstrated on one extra environment, i.e. it is hard to assess how significant it is to drive for instance 42m further than another method (+/- 52m) to take the first line of Table 1 as an example.

**Questions:**

- Why would the policy benefit from all the fusion in the value function if it is a neural network only taking the global modality feature as input?

- From the results in Fig 5 it seems bulk of the contribution of the value is from the RGB frame, as its weight never goes below 0.75?

- The only difference between the proposed approach and the baselines is the value function estimation. I would wonder whether the architecture yields a better value function (i.e. closer to the ground truth value) faster, which yields the improved training of the policy. However, if in the long run, any value iteration converges to the true value function, it should yield the same policy, no? I would be interesting to inspect the MSE of the value function compared a ground truth value estimate for the different algorithms over time, to see if this is indeed the case.

**Limitations:**

The authors only mention the limitation to one set of modalities in their work. I would make this broader to note that the approach is only tested on this one environment.

---

> ### Author Rebuttal · Authors · 2023-08-10
>
> We are truly thankful for the thoughtful remarks and the experimental recommendations you have provided. These suggestions shed light on what we can improve, and we believe they will be instrumental in refining our work further. We address your main concerns as follows:
>
> > ...fusing the q-values is complex. Why would the policy benefit from this if it only takes global modality feature as input?
>
> Thanks for the great question. First, we wish to emphasize: it is the value function that provides supervision signals for the policy network. Therefore, the factors that influence the policy network are two-fold: 1) accurate value estimation, and 2) robust input features. Our paper primarily addresses the first, using multi-modal strategies for better Q-value estimations. When the Q-value function is accurate, it enriches the feedback to the policy by reliably estimating future rewards for potential actions. This is the reason behind our hierarchical design, which balances individual modality contributions for optimal Q-values.
>
>
> As for the input features, our work shows that the robust Q-values estimated by HAVE can already drive the policy network to select predictive features from only the global modality feature. While we could introduce a complex modality fusion module, its success still hinges on precise Q-values, which drive system learning. Thus, we prioritized Q-value estimation. However, your question raises a novel and natural extension of our work: to develop proper feature selection mechanisms for the policy network. We sincerely appreciate this and will definitely work on it in the future.
>
>
> > ...very high spread on the results. How significant is the improvement?  ... seems the other methods are not necessarily converged yet.
>
> Thanks for highlighting this. Result variance is due to initialization randomness, a known RL challenge (cf. experiments in [1],[2]). We averaged results from five runs with different seeds for Fig. 3 and Fig. 4 following common practice [1][2]. The results reveals HAVE's superior performance by 120K steps. Some curves might not seem converged at 120K. With limited computation resources during rebuttal, we re-run experiments on the most questionable non-converged figures (i.e. Fig. 3 (left) and Fig. 4 (bottom left) ), each with the two most suspicious non-converged methods. The results are given in Fig.2 (b) and Fig.2 (c) in the rebuttal PDF, respectively. These results show convergence around 150K steps, with HAVE still leading. In future revisions, we'll extend all experiments to 150K steps.
>
> > It would be more convincing if the approach was demonstrated on one extra environment.
>
> Thanks for your valuable suggestion.  We have conducted more experiments on an additional environment DeepMind Control Suite for robot control. Please refer to our general response posted above for detailed results. Experiment results show that HAVE also works positively in the new environment and can improve over other methods.
>
> > ...the contribution of the value from the RGB frame...never goes below 0.75?
>
> Thanks for this keen observation. A higher weight to RGB signals over event signals can be attributed to two factors:
>
>
> 1) Content richness. Compared to events, RGB images contain substantially more information, especially static ones like colors and textures. Such information is important for decision-making. Event signals, on the other hand, mostly reflect edges that are moving. In our demonstration, the RGB frames are clearly discernible by the agent. As a result, they dominate the modality contribution with abundant content. In Fig.2(a) of the rebuttal PDF, we provide the modality weight under extremely low-light conditions where the RGB frames are barely observable, we see the event signals play a more important role in such situations.
>
>
> 2) Event signals are derived from discrete intensity shifts, which can introduce temporal noise and affect data reliability. Meanwhile, RGB signals offer a smoother, more consistent environment depiction, reducing noise concerns.
>
> > Whether the architecture yields a better value function faster, I would be interesting to inspect the MSE of the value function compared a ground truth value estimate for the different algorithms over time.
>
> We provide the MSE curve in the ClearNight weather of CARLA, please see Fig.2(e) in the rebuttal PDF. The ''ground truth'' depicted in this figure is computed using the Monte Carlo method, which calculates the cumulative reward from the current state to the end of the test episode. From Fig.2(e) we see that although all methods tend to reduce their value error. Their MSE is closer along training, yet HAVE still outperforms all methods at the end of training. It is notable that such a ground truth value can not be obtained beforehand during training, so this is just for reference and the rigorous analysis needs to further explore.
>
> RL always uses Temporal Difference (TD) learning to update network, where the learning goal of value functions is updated by collective rewards and current value estimators (i.e., critic network). That is, there is no real ground truths for Q value during training.  We cannot train a good value estimator just by a long-time training because due to accumulate errors. Hence, it is very important to design a better value estimator, which is our main goal.
>
> **Reference**
>
>
> [1] Soft Actor-Critic: Off-Policy Maximum Entropy Deep Reinforcement Learning with a Stochastic Actor, ICML 2018.
>
>
> [2] Image augmentation is all you need: Regularizing deep reinforcement learning from pixels, ICLR 2020.

---

> > ### Comment · Reviewer_9NFJ · 2023-08-11
> >
> > I thank the authors for their responses, and I appreciate the additional results provided. I have adjusted my score accordingly.

---

> > > ### Author Response · Authors · 2023-08-11
> > > **Thank you for your positive support**
> > >
> > > We sincerely appreciate your experimental advice and inspiring questions that give us new research directions. We have learned a lot from the review. Thank you again for your effort and supportive feedback.

---

### Official Review · Reviewer_VLp8 · 2023-07-08

**Soundness:** 3 good
**Presentation:** 3 good
**Contribution:** 3 good
**Rating:** 6
**Confidence:** 4

**Summary:**

The paper proposes a Hierarchical adaptive value estimation (HAVE) framework for multi-modal (RGB, event, depth) visual reinforcement learning. Firstly, a modality-specific value function learning process is proposed, and an assignment module is proposed to weight different modality. Second, a re-fusion is proposed to futher combine global feature based value and modality-specific based value. Method is evaluated on CARLA simulation environment. Various baselines, including SAC, DrQ, DeepMDP, etc are compared and the results show the proposed HAVE perform better under different weather conditions.

**Strengths:**

The decomposition of global value function into weighted combination of modality value function is novel and well-motivated by the observation that simple feature fusion might cause dominant modality behaviour. The weight assignment module is well designed and also show meaningful behaviour in Figure 5. The re-fuse further makes the method can enjoy both the value and feature fusion, achieving better results. Results are strong compared to other methods on CARLA across different weather conditions

**Weaknesses:**

1. The auxiliary loss term in equation 14 is not ablated.
2. Ablation study on the design of re-fusion is missing. Why such dynamic fusion (i.e. predict FC weights and bias) is necessary, will other simple way of fusion work?
3. The introduce of weight assignment and re-fusion leads to increase of #params and #FLOPS, which is not listed in the paper. And further, to show the performance improvement is not coming from increase of #param and FLOPs, ablation study can be designed and conducted to compare baselines with similar #params and FLOPS.


**Questions:**

Can authors address the concerns in weaknesses part

**Limitations:**

As discussed by authors, currently one RGB, event, Depth modalities are considered, other modalities remains unknown. And currently the method is only evaluated on one driving simulation environment, whether it can be applied to other environment or real world is unknown.

---

> ### Author Rebuttal · Authors · 2023-08-10
>
> First and foremost, we would like to express our gratitude for your positive comments. We also deeply appreciate the practical experimental suggestions given in your constructive feedback, which offers valuable perspectives that will strengthen our work. Below are our responses to your concerns:
>
> >1. The auxiliary loss term in equation 14 is not ablated.
>
> We agree that conducting an ablation study of auxiliary losses would elucidate their contributions. We have carried out the corresponding experiments in the ClearNight environment of CARLA. The results are illustrated in Fig.2(d) of the rebuttal PDF. Specifically, three terms in Eq.13 comprise $L_{aux}$: the global transition loss, the sum of individual transition losses, and the reward prediction loss. When compared to the full results of HAVE in Fig.4 (top left) of the main paper, Fig.2(d) reveals that omitting any of these terms negatively impacts model performance. Among them, the reward prediction loss is the most crucial, leading to the most significant performance drop. When all three terms are excluded (i.e., the 'w/o aux' curve in Fig.2(d)), there is a notable decline in performance.
>
>
> >2. Ablation study on the design of re-fusion is missing. Why dynamic fusion is necessary, will other simple way of fusion work?
>
> Thanks for your good question. This question can be separated into two parts.
>
> On why we need dynamic fusion over static ones (such as taking the mean of GVE and LVE values), since LVE does not always outperform when GVE provides accurate estimates, and conversely, GVE does not always excel when LVE estimates are precise. Their effectiveness is contingent upon the attributes of the current global modality, which can vary drastically depending on the environmental context. To address this variability, we employ a hypernetwork for a dynamic combination of both methods. While we have experimented with directly merging GVE and LVE, Table 1 in the rebuttal PDF indicates that, although direct fusion is feasible, the dynamic fusion through the hypernetwork proves more efficacious.
>
> On why we need a hypernetwork to perform dynamic fusion, the rationale behind employing hypernetworks is to avoid directly putting the global modality feature $f_t^g$ into the mixing network. This approach is taken because ultimately, we want the aggregated value benefits from the additional global modality information in $f_t^g$ but not overly influenced by it. If we directly put $f_t^g$ into the mixing network, the output value will arithmetically depend on it. This could impose unnecessary fusion difficulty since $f_t^g$ may vary significantly across different environment situations. Meanwhile, the re-fusion process is not strictly a task-level value fusion anymore because the feature $f_t^g$ has actively engaged in the fusion process. Instead, the utilization of hypernetworks offers the ability to condition the mixing network's weights on $f_t^g$ in a versatile manner while not overly constraining the fusion process.
>
> Hypernetworks generate weights for primary networks, allowing for dynamic adaptation, parameter efficiency, and conditional computations, making them versatile tools for various tasks[1,2,3].
>
> [1] A Brief Review of Hypernetworks in Deep Learning. arXiv preprint arXiv:2306.06955 (2023).
>
> [2] Principled weight initialization for hypernetworks. ICLR. 2019.
>
> [3] Continual learning with hypernetworks. ICLR. 2020.
>
>
> >3. Increased #params and #FLOPS are not listed in the paper. Ablation studies should be designed to compare baselines with similar #params and FLOPS.
>
>
> Thanks for your suggestion. A comparison of baselines with a similar model complexity is indeed instructive. To address this, in the rebuttal PDF's Table 2, we outline the number of parameters (#params) and FLOPS for HAVE, TransFuser, and EFNet. A few key observations emerge:
>
> 1. All three methods share the same observation encoder architecture.
> 2. As a value estimation method, HAVE directly concatenates modality features and doesn't incorporate advanced feature fusion modules. Conversely, TransFuser and EFNet introduce significant #params and FLOPS with their attention-based modality feature fusion modules.
> 3. The critic network in HAVE includes weight assignment and re-fusion networks, adding to its complexity. Nonetheless, this added complexity is modest when juxtaposed with the feature fusion modules of TransFuser and EFNet. Overall, HAVE still achieves the best result with the least #params and FLOPS. It is also worth noting that the critic is utilized only during the training phase and not used in testing. Thus, there is no actual increase in either the parameter counts or FLOPS for HAVE at the test stage.

---

> > ### Comment · Reviewer_VLp8 · 2023-08-20
> > **Thank authors for the response**
> >
> > Thank authors for the response, the detailed responses have resolved most of my concerns.

---

> > > ### Author Response · Authors · 2023-08-21
> > > **Thank you for your feedback**
> > >
> > > We deeply appreciate the time and effort dedicated to your detailed and informative review. We will incorporate the experimental advice in our revised manuscript. Thank you again for your support and valuing of our work.

---

### Author Rebuttal · Authors · 2023-08-10

**To All Reviewers (Additional Experiments on New Environment)**

We thank all the reviewers for their valuable time and insightful feedback. In this general response, we would like to address the concerns about the effectiveness of our HAVE in environments other than autonomous driving. Specifically, we conduct additional experiments using the Mujoco-powered **DeepMind Control Suite** [1] with RGB and depth as input modalities on two challenging tasks (Cheetah, run and Walker, walk). The training and testing curves of HAVE and other comparable methods are given in Fig.1 in the uploaded one-page rebuttal PDF. From the results, it is clear that HAVE also outperforms other methods in these robot control tasks.

**Reference**:

[1] dm_control: Software and tasks for continuous control, Software Impacts, 2020

---

### Decision · Program_Chairs · 2023-09-21

**Decision:**

Accept (poster)

**Comment:**

The paper offers a novel approach to multi-modal reinforcement learning and addresses a significant research area with practical implications. The quality of writing, presentation, and language in the paper is commendable. The authors' commitment to addressing reviewers' feedback through additional experiments and plans for future research demonstrates their dedication to improving the paper.

The AC thinks indeed there are similarities between the attention based combination of value functions in multi-agent RL (Q-Mix) and the idea presented in the paper. But the authors adequately explained this concern in their rebuttal: "a similarity between the roles of “agent'' in multi-agent RL and “modality'' in multi-modal RL: both exhibit distinctive contributions to the final reward and require effective cooperation to optimize the globaldecision. This resemblance highlights a shared principle across diff erent domains within RL, which by itself isa new kind of discovery not mentioned by existing works." The AC thinks the observation that a similar idea of cooperation between different multi-modal features and multi-agent features is novel and interesting.

Regarding the lack of string baselines, the authors also added another baseline in rebuttal.

Given the overall response from reviewers and the authors' proactive approach to addressing concerns, I recommend accepting this paper.